# Offshore transport of particulate organic carbon in the California Current System by mesoscale eddies

Caitlin M. Amos [1], Renato M. Castelao[1]* & Patricia M. Medeiros[1]

The California Current System is characterized by upwelling and rich mesoscale eddy activity. Cyclonic eddies generally pinch off from meanders in the California Current, potentially trapping upwelled water along the coast and transporting it offshore. Here, we use satellite-derived measurements of particulate organic carbon (POC) as a tracer of coastal water to show that cyclones located offshore that were generated near the coast contain higher carbon concentrations in their interior than cyclones of the same amplitude generated off-shore. This indicates that eddies are in fact trapping and transporting coastal water offshore, resulting in an offshore POC enrichment of $20.9 \pm 11$ Gg year$^{-1}$. This POC enrichment due to the coastally-generated eddies extends for 1000 km from shore. This analysis provides large-scale observational-based evidence that eddies play a quantitatively important role in the offshore transport of coastal water, substantially widening the area influenced by highly productive upwelled waters in the California Current System.

[1] Department of Marine Sciences, University of Georgia, Marine Sciences Building, 325 Sanford Drive, Athens, GA 30602, USA. *email: castelao@uga.edu

Mesoscale eddies with radius on the order of 100 km are ubiquitous features[1] known to influence the horizontal and vertical distribution of physical and biogeochemical properties throughout the global ocean[2–7]. The majority of these eddies are nonlinear (ratio of rotational speed to translational speed >1), meaning they can theoretically trap water parcels and associated properties during formation[1]. Eddies can then potentially transport the trapped properties for hundreds of kilometers throughout the ocean[8,9].

In the California Current System (CCS), a highly productive Eastern Boundary Current System (EBCS)[10], eddies are commonly observed[1,11,12] and are thought to play a role in redistributing nutrients into the oligotrophic, offshore region[13–15]. Persistent equatorward winds during the summer produce offshore surface Ekman transport and upwelling along the coast, bringing cold, nutrient-rich waters to the surface. This results in a band of elevated nutrients and particulate organic carbon (POC) concentrations near the coast[10,16–19]. As the California Current flows southward along the coast, meanders in the current produce filaments that can pinch off as cyclonic (counterclockwise in the Northern Hemisphere) and anticyclonic (clockwise) eddies that then propagate westward[7,12,14,15]. During formation of the eddies, the nutrients and properties associated with the upwelled coastal water could be potentially trapped in their interior[20]. Modeling studies suggest that eddies can transport the trapped coastal water offshore, contributing to the redistribution of carbon, nutrients, and other properties in the CCS[13,15,21,22] and in other upwelling systems[23]. While previous observational studies have shown individual examples of eddies transporting materials in various regions of the ocean[9,20,24,25], no large-scale observational study has systematically quantified eddy-induced transport and its influence on carbon distribution in highly productive upwelling systems. The offshore transport of coastal water that is rich in carbon and nutrients in the CCS could have important implications for the marine ecosystem. Therefore, quantifying the eddy-induced transport by using observations is important for further understanding the influence of eddies in EBCS. By using satellite-derived measurements of POC as a tracer of coastal water, we show that cyclonic eddies located offshore that were generated near the coast contain higher carbon concentrations in their interior than cyclonic eddies of the same amplitude generated locally offshore, contributing to the enrichment of POC in the offshore region.

## Results

**Effects of eddies on POC distribution.** Developments in eddy detection and tracking algorithms by using altimetry data have made it possible to study mesoscale eddies and their properties over large spatial areas and long-time periods[1,26,27]. Here, we use 13 years of satellite-derived measurements of POC[28] as a tracer of coastal water to show that eddies can play an important role in redistributing carbon from the coastal region to offshore areas in the CCS (Fig. 1). Meanders and filaments[29] extending westward from the California Current are often distinguished by elevated POC concentrations as they initiate the offshore transport of coastal water (Fig. 1a). A filament can pinch off to the west of the current as a cyclonic eddy (Fig. 1b, c), entraining upwelled coastal water that is rich in carbon and nutrients from the shoreward side of the current. The trapped coastal water then gets transported offshore for hundreds of kilometers, as indicated by the elevated POC concentrations remaining months later in the interior of the eddy compared with the surrounding offshore waters (Fig. 1d).

The fraction of surface POC in the CCS that is found in the interior of eddies is calculated from the 13 years of satellite-derived POC by using the locations and sizes of eddies identified in this region from an existing global eddy dataset[1]. In the region 300–1200 km from the coast between 33° and 43°N, which is outside of the 300-km nearshore band where POC concentrations are the highest, about 6.9% of the total amount of POC can be found inside cyclonic eddies that are generated in this region, and these eddies occupy about 7% of the total area (Fig. 2a). In contrast, 13.4% of the total amount of POC is found inside cyclones generated inshore of 300 km and propagated to the offshore region, but these eddies only occupy 10.4% of the total offshore area during September–February (Fig. 2c). The relative enrichment of POC content in the interior of cyclones that were generated near the coast and propagated offshore compared with the area occupied by those eddies in the offshore region is the largest from late summer to early winter (Fig. 2c). This is consistent with eddies being generated near the coast during late spring/early summer (May–August) and propagating offshore at ~2 km day$^{-1}$ [12,14,26,27], reaching the offshore region 3–6 months later.

Although anticyclonic eddies are also capable of trapping and transporting materials, they have less of an impact on the redistribution of recently upwelled, carbon- and nutrient-rich water in the CCS compared with cyclones due to differences in the water that is entrained during formation. Anticyclonic eddies are often generated from meanders that pinch off from the shoreward side of the California Current[15,20], trapping oligotrophic offshore water that contains lower concentrations of POC and nutrients than the water closer to the coast that is trapped by cyclones (as shown in Fig. 1). Because of this, anticyclones formed inshore of 300 km from the coast that propagated offshore

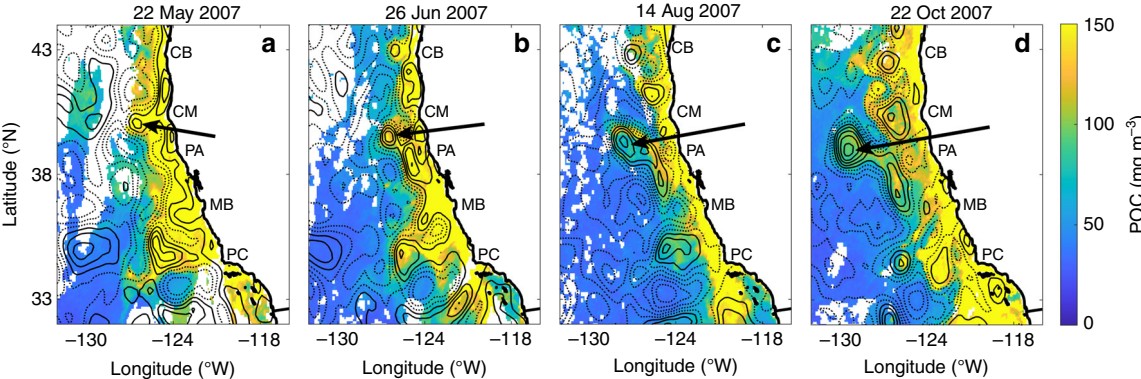

**Fig. 1** Example of offshore transport by an eddy in the California Current System. Particulate organic carbon (mg m$^{-3}$) derived from satellite observations and mean sea-level anomaly contours from altimetry at 4-cm intervals on **a** 22 May 2007, **b** 26 June 2007, **c** 14 August 2007, and **d** 22 October 2007. Solid contours are negative. Arrows mark the location of a cyclonic eddy transporting coastal water rich in carbon offshore. CB Cape Blanco, CM Cape Mendocino, PA Point Arena, MB Monterey Bay, PC Point Conception

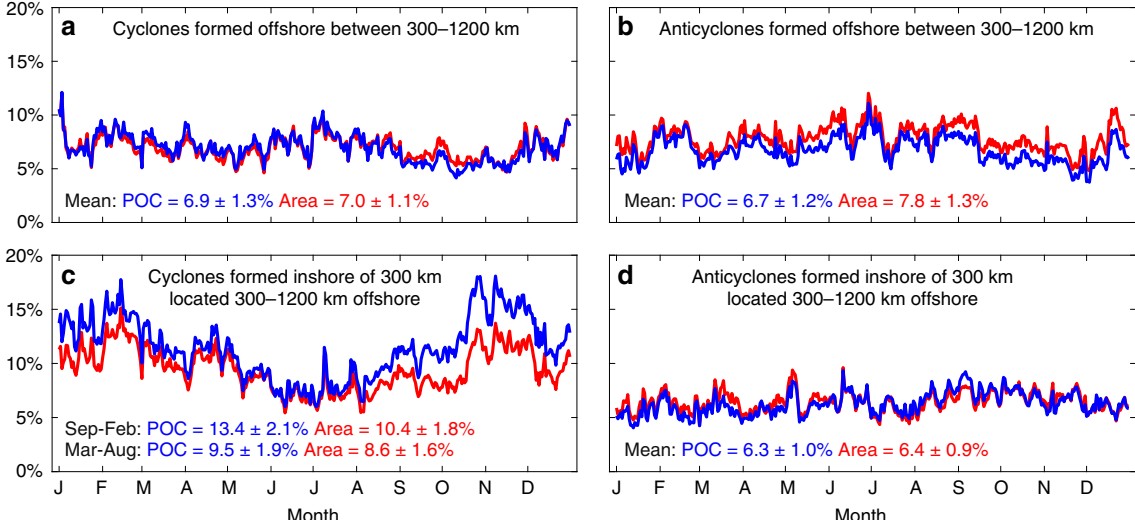

**Fig. 2** Carbon content and spatial area occupied by cyclonic and anticyclonic eddies (1997–2010). Percentage of the total amount of surface particulate organic carbon (blue line) and the total area (red line) between 300 and 1200 km from the coast that is inside all eddies for **a** cyclones and **b** anticyclones formed offshore between 300 and 1200 km from the coast and **c** cyclones and **d** anticyclones formed inshore of 300 km and located between 300 and 1200 km offshore

contained lower percentages of the total POC in the offshore region compared with cyclones (Fig. 2b, d), further indicating that anticyclones are trapping water with less POC, nutrients, and other properties associated with the upwelled coastal water when they form. Anticyclonic eddies therefore do not contribute substantially to the offshore transport of coastal water like their cyclonic counterparts.

**Lateral transport of coastal tracers by cyclonic eddies**. To detect eddies transporting trapped coastal water via satellite observations, the anomalous POC signature associated with each eddy was isolated and compared between cyclones generated inshore and offshore of 300 km from the coast (see Methods and Supplementary Figs. 1 and 2). Cyclonic eddies in the CCS are associated with positive POC anomalies (Fig. 3a–c). The concentration of POC in the eddies' interior is generally highest for cyclones generated and located inshore of 300 km from the coast (Fig. 3a, d). Except for eddies of small amplitude (3–5 cm), cyclones that are generated inshore and propagate offshore of 300 km from the coast have higher POC anomalies than cyclones of the same amplitude generated offshore between 300 and 600 km (Fig. 3b–d). The POC enrichment in the cyclones generated inshore and located offshore in comparison with those generated offshore could be due to the offshore transport of the POC that was trapped during eddy formation near the coast, remineralization of the trapped POC and recycling into new carbon, and local production as the eddy propagates offshore through the utilization of nutrients that were trapped at formation. All these sources of POC are influenced by the trapping of POC- and nutrient-rich coastal upwelled water by eddies during formation and subsequent offshore lateral transport. On average, it takes cyclonic eddies about 4–6 months to reach 300–600 km offshore. Extending the analyses farther offshore in additional 300-km-width bands reveals that the POC signature associated with cyclonic eddies generated inshore that propagated offshore remains higher than the signature for cyclones generated in the offshore region in the 600–900 and the 900–1200 km bands from the coast (Fig. 4). No significant difference in POC content is observed offshore of 1200 km from the coast, indicating that POC

enrichment in offshore waters due to the influence of coastally generated eddies is most important within about 1000 km from shore. For both cyclonic eddies generated inshore or offshore of 300 km from the coast, the mean POC anomaly decreases as the eddies propagate westward (Fig. 4).

For cyclonic eddies formed and located inshore of 300 km from the coast, the monthly mean POC anomaly is the highest during May–July (Fig. 5a), following a seasonal pattern that is consistent with the seasonality of eddy generation[14,26,27] and upwelling[16,18] along the coast in the CCS. About 55% of cyclonic eddies formed inshore of 300 km from the coast propagate to the 300–600-km offshore region (Supplementary Table 1). The mean POC anomaly associated with these eddy peaks between November and January (Fig. 5b), about 4–6 months after the peak in the POC anomaly associated with the eddies is observed when they were located <300 km offshore (Fig. 5a). The distance that the eddies that are generated inshore travel to the offshore region, and the delay in the timing of the peak POC anomaly when the eddies are located inshore and offshore of 300 km, indicate propagation speeds of ~2 km day$^{-1}$, which is consistent with the known eddy propagation speeds in the CCS[14,26,27]. This consistency in eddy propagation speed and the time difference in the peak in POC anomaly between cyclones located inshore and offshore provides further evidence that cyclones are indeed transporting coastal water offshore. Cyclonic eddies generated offshore between 300 and 600 km show no clear seasonal trend in the POC anomalies, except for a small decrease during summer (Fig. 5c).

The mean volume transport by cyclones can be calculated by using the average volume associated with the cyclonic eddy occurrences of different amplitudes that are generated near the coast and then propagate offshore. We focus on cyclones because of their larger influence on the offshore transport of coastal water that is influenced by upwelling (Fig. 2a, c) compared with anticyclones (Fig. 2b, d). In the CCS, on average, $6.0 \pm 0.6$ cyclonic eddies with lifetimes longer than 4 weeks are generated inshore of 300 km from the coast and propagate offshore of 300 km annually between 33° and 43°N (Fig. 6a)[1]. Considering an average trapping depth inside cyclones of 400 m that was estimated from our modeling results by using large cyclonic eddies that are comparable to the satellite-detected eddies and is

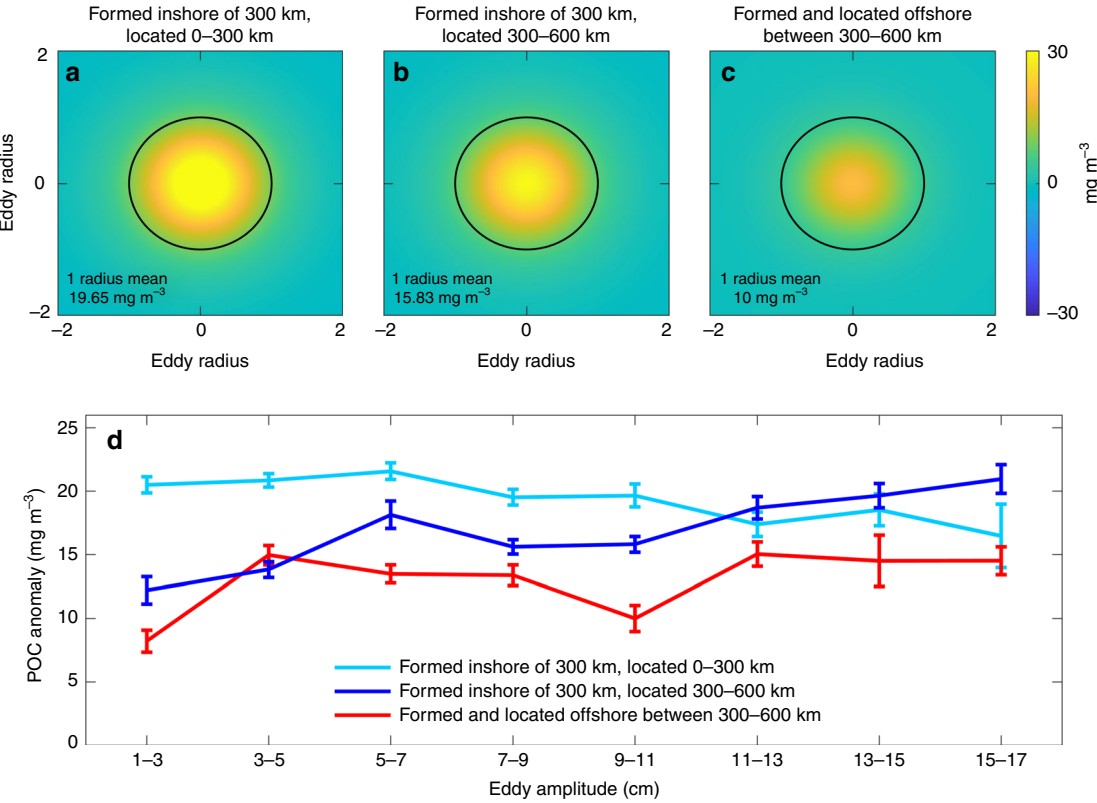

**Fig. 3** Particulate organic carbon composites and anomalies. Cyclonic eddies identified from satellite altimetry were collocated with particulate organic carbon (POC) anomalies that were derived from satellite observations. To isolate the POC signature associated with each eddy, the POC anomaly was extracted within 2-by-2 eddy radii from the eddy center, and the distance from the eddy center was normalized by the eddy radius (unitless). A 2D Gaussian function was fitted to each POC anomaly field, and composites were computed by averaging the POC anomalies for eddies of similar amplitude (see Supplementary Fig. 2). **a–c** Composites of POC anomalies within 2-by-2 eddy radii for cyclonic eddies with amplitudes 9–11 cm. The black circle represents the boundary for one eddy radius. **d** Mean POC anomaly (mg m$^{-3}$) and standard error within one eddy radius grouped by eddy amplitude

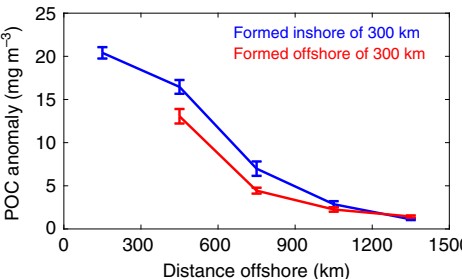

**Fig. 4** Particulate organic carbon anomaly as a function of distance from the coast. Mean particulate organic carbon anomaly (mg m$^{-3}$) and standard error within one eddy radius for cyclonic eddies formed inshore of 300 km from the coast (blue line) and for those formed offshore of 300 km (red line) as the eddies propagate offshore

consistent with previous studies[14], our analyses indicate that 65 ± 7% of the volume of water initially trapped inside cyclones remains trapped after the eddies propagate offshore (see Methods and Supplementary Fig. 3). By using this trapping efficiency, the resulting annual volume transport in the top 400 m by cyclonic eddies in the CCS is $1.06 ± 0.2$ Sv ($1$ Sv $= 10^6$ m$^3$ s$^{-1}$; Fig. 6b; see Eq. (2) in methods). To provide error bounds for this estimate, using the ±7% error in the trapping efficiency results in volume transports of 0.95 and 1.2 Sv. This range of values for the volume transport is comparable to the estimated annual mean integrated transport due to eddies in the Canary Current System (1.3 Sv in

the top 300 m)[30]. The relationship between in situ POC at the surface and integrated from the surface to 100-m depth[31] (see Methods and Supplementary Fig. 4) can also be used to estimate the amount of POC that is added to the offshore region by cyclonic eddies generated near the coast that are trapping the upwelled coastal water and transporting it offshore (Supplementary Fig. 5). For that, the differences in the POC content between cyclonic eddies located 300–600 km offshore that were formed inshore of 300 km and those formed offshore between 300 and 600 km from the coast are used (Equation 1; see also Supplementary Fig. 5 and Fig. 3b–c). Lateral transport by cyclonic eddies results in a POC enrichment offshore of 20.9 ± 11 Gg year$^{-1}$ in the top 100 m (Fig. 6c). As mentioned before, this POC enrichment may be associated with the direct lateral transport of POC from the coastal region that was trapped in the interior of cyclonic eddies during formation, but it could also be related to the offshore transport of nutrient-rich water that is also trapped inside the eddies, which supports production, or to the trapped POC being locally remineralized and recycled into new carbon. All these sources of POC are resultant of coastal water being trapped by cyclonic eddies and transported offshore. We note that changes in the mixed-layer depth[32] or the euphotic zone depth may result in different amounts of POC being trapped inside the eddies. If the calculations are repeated for shallower layers of 20 or 50 m, the enhancement of POC in the offshore region is estimated at 5.4 ± 2.9 and 12.2 ± 6.1 Gg year$^{-1}$, respectively. We note that the correlation coefficient between surface POC and integrated POC for different depth ranges is approximately depth independent in the top 100 m.

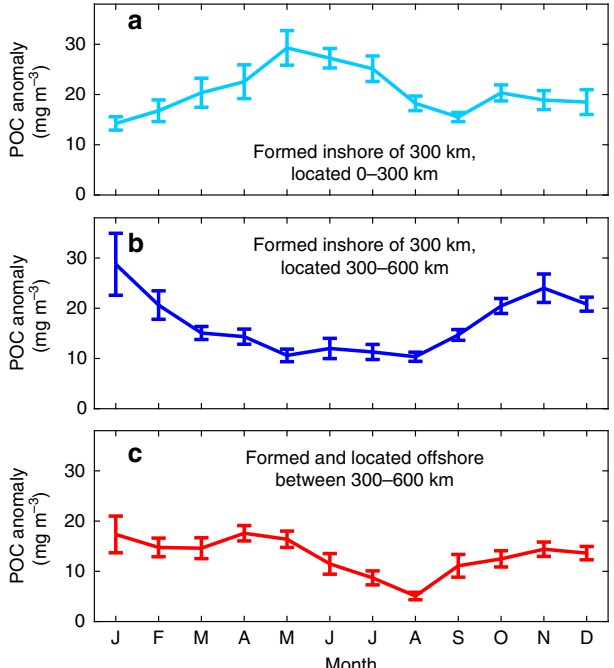

**Fig. 5** Monthly particulate organic carbon anomaly. Mean particulate organic carbon anomaly (mg m$^{-3}$) and standard error within one eddy radius as a function of the month of the eddy occurrence for cyclonic eddies **a** formed and located inshore of 300 km, **b** formed inshore of 300 km and propagated to 300–600 km offshore, and **c** formed and located offshore between 300 and 600 km

## Discussion

Previous observational studies have shown examples of mesoscale eddies transporting properties in different regions of the ocean[9,24,25]. Modeling studies have also suggested that eddies can trap and transport materials offshore in the CCS[15,22] and in other EBCS[23], resulting in reduced biological production in the near-shore environment[13]. In this study, we present the first observational evidence that this process is important enough to produce a systematic signature in eddies far from the coast in the CCS. Our results indicate that cyclonic eddies formed near the coast are capable of trapping carbon- and nutrient-rich coastal water and transporting it offshore for hundreds of kilometers. Cross-shelf transport in the CCS is important for increasing the area influenced by highly productive upwelled waters. Studies have shown that the offshore deflection of the surface-intensified upwelling jet and its associated meanders and filaments extends about 400 km offshore on average[12,19,33], inducing cross-shelf transport of 1–2 Sv[34–37]. These filaments can transport organic carbon offshore in a very intense but coastally confined manner, dominating the mesoscale offshore transport in EBCS within 500 km from shore and contributing to up to 80% of the total flux of organic carbon at 100 km offshore[23]. Our novel observational-based study reveals that the signature of the enhanced POC in the interior of cyclonic eddies generated near the coast is detectable until about 1000 km from shore, indicating the role of eddies in redistributing POC and coastal water across a wider area. We estimate that cyclonic eddies transport ~1 Sv offshore, indicating that this mechanism can be just as important as the offshore deflection of the upwelling jet[34]. Furthermore, the enrichment of POC in the offshore region induced by cyclonic eddies generated near the coast is estimated to be about 20.9 ± 11 Gg year$^{-1}$, with about 70% occurring between late summer and early winter. The annual POC enrichment offshore due to cyclonic eddies can be as high as 30–35% of the total amount of POC introduced into the

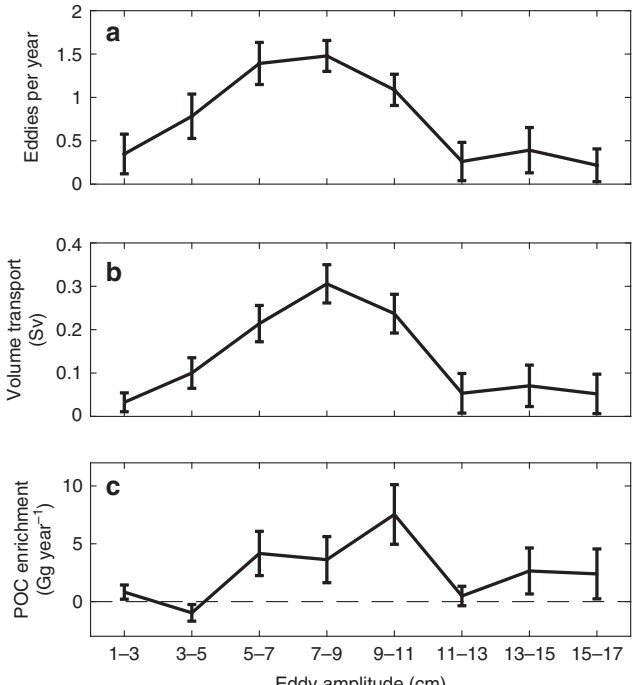

**Fig. 6** Estimates of lateral transport by cyclonic eddies in the California Current System. **a** Average number of cyclones with lifetime longer than 4 weeks in the eddy dataset[1] formed inshore of 300 km from the coast that propagate to at least 300 km offshore per year grouped by average amplitude of the eddies when they are located offshore of 300 km. Total number of cyclonic eddies propagating offshore per year is 6.0 ± 0.6. Error bars represent the standard error. **b** Mean volume transport (Sv) and standard error in the top 400 m by cyclonic eddies calculated assuming that 65 ± 7% of the initially trapped volume of water is transported offshore (see Methods and Supplementary Fig. 3). Total transport summed across the different amplitude bins is 1.06 ± 0.2 Sv. **c** Particulate organic carbon enrichment (Gg year$^{-1}$) and standard error in the offshore region due to cyclonic eddies generated inshore of 300 km from the coast that propagate offshore (see Supplementary Fig. 5). Total particulate organic carbon enrichment summed across the different amplitude bins is 20.9 ± 11 Gg year$^{-1}$

CCS by the Columbia River (USA) annually[38–40]. Total carbon redistributed by cyclonic eddies is likely to be significantly larger since our estimate only includes the particulate phase, and dissolved organic carbon concentrations can be 10–25 times larger than POC content in near-surface waters in the CCS[41,42]. In addition to carbon and nutrients, the trapped coastal water contains other materials that are also presumably being redistributed to the offshore region by cyclonic eddies.

The estimates of volume transport and POC enrichment are influenced by the limitations associated with detecting the eddies by using satellite observations. Imperfections in the eddy detection and tracking algorithms may result in distortions in the identified eddies, especially when eddies are interacting with other eddies or other mesoscale features[43]. Uncertainties in eddy characteristics, such as radius and amplitude, could affect the transport calculations. In addition, only eddies with radii larger than 40–50 km are detectable with altimetry data[1]. The processing of the satellite-derived POC observations, specifically the use of a Gaussian fit to isolate the eddy signature, also results in smoothed fields by removing small-scale variability. Since sub-mesoscale eddies are also abundant in the CCS and may further contribute to offshore transport and subduction of materials[44–46], it will be important for future studies to further understand the

role of these smaller eddies and to quantify their relative importance on offshore transport in the CCS.

The CCS is one of four major EBCS, all of which share similarities in eddy activity[11,26,47]. The eddy-driven offshore transport of coastal water will presumably be important in the other EBCS as well. Eddy activity, mesoscale variability, and upwelling in EBCS are linked to winds[12], which are likely to change in the future[48–51]. Therefore, changes in the seasonality of mesoscale activity and eddy generation are also possible. Our analyses suggest that changes in eddy activity would likely result in changes in offshore transport of coastal water that is rich in carbon and nutrients, and this could have important implications for the marine ecosystem in highly productive EBCS.

## Methods

**Mesoscale eddies.** The location and characteristics of mesoscale eddies in the CCS used in this study were obtained from the fourth release of an existing global dataset of mesoscale eddies[1] (wombat.coas.oregonstate.edu/eddies/index.html). To detect mesoscale eddies, daily sea level anomaly (SLA) fields produced by Archiving, Validation, and Interpretation of Satellite Oceanographic data (AVISO) are first spatially filtered to remove large-scale variability. Mesoscale eddies in the fourth release of the dataset are detected by using a method that grows eddies from individual SLA extrema[52]. The growing method starts with individual SLA extrema (positive for anticyclones and negative for cyclones) and finds all neighboring pixels whose SLA values lie above a sequence of thresholds. An eddy is defined when the set of connected pixels satisfies a set of criteria used to define compact and coherent structures. Eddies are then tracked by pairing eddy realizations that are within allowable ranges of amplitude, radius, and distance of the initial eddy at subsequent time steps. The eddy detection and tracking algorithms are described in detail in refs. [1,52]. Given the resolution of the AVISO satellite fields, only mesoscale eddies with radius larger than ~40–50 km are resolved; therefore, submesoscale and smaller mesoscale variability are not included in the dataset and in the analyses. Also, complications can arise when eddies merge or interact with other eddies or from noisiness in the SLA fields. This can result in imperfections in the detection of the boundaries and characteristics (e.g., radius and amplitude) of the eddies[1,43]. Despite these limitations, animations of the eddy tracks on SLA fields indicated that the dataset captures most large mesoscale eddies in the CCS.

Nonlinear eddies located in the CCS between 33°−43°N and 0–1500 km from the coast during the time period of the satellite POC data (1997–2010) were identified from the dataset for the analyses. Eddies are considered nonlinear if the ratio $U/c > 1$, where $U$ is the maximum rotational speed and $c$ is the translation speed of the eddy estimated at each point along the trajectory[1]. In total, 553 cyclonic eddy tracks were studied. Calculations of the nonlinearity parameters indicated that the majority of the eddies were nonlinear for at least 80% of their lifetime. The distance between the coastline and the location of the eddy centers at each point along the eddy trajectories was calculated to distinguish eddies generated or located inshore and offshore of 300 km from the coast (Supplementary Fig. 1). The threshold of 300 km from the coast was chosen to define inshore and offshore based on the average width of the band of high POC concentrations along the coast. The band of high POC extends about 250 km from the coast on average. Since POC is used here as a tracer of coastal water, a distance larger than the average width of the band with high POC was chosen to distinguish between upwelled coastal water and offshore water. The distance of 300 km is also consistent with the average width of the coastal band with high sea surface temperature frontal activity and of the meandering upwelling jet[53]. To check that the results are not sensitive to the distance threshold, the analyses were repeated by using other distances from the coast, e.g., 350 km, and the results were consistent to those presented here. The offshore region was divided into bands of 300 km in width for the analyses (300–600, 600–900, 900–1200, and 1200–1500 km from the coast). The width of the offshore bands was chosen to be the same as the inshore region (0–300 km).

**POC measurements.** Daily remote-sensing reflectance data from Sea-Viewing Wide Field-of-View Sensor (SeaWiFS; oceandata.sci.gsfc.nasa.gov/SeaWiFS/) were used to estimate POC concentrations[28]. Data are available daily from September 1997 to December 2010 at 9-km resolution. To reduce the influence of cloud coverage, data were averaged at a 7-day interval (Supplementary Fig. 2a). The mesoscale structures that are of primary interest here are obscured by the large-scale POC background distribution. Spatial high-pass filtering[52,54] the weekly POC fields (6° longitude by 6° latitude window) to remove the large-scale patterns allowed for isolating the POC anomaly associated with mesoscale activity in the region[6,20,25] (Supplementary Fig. 2b). Quantitatively similar results are obtained if the large-scale patterns in the region were removed by computing the long-term weekly averaged POC distribution, instead of using a spatial filter. Cyclonic eddies, which were identified by using altimetry data[1] (black box in Supplementary Fig. 2b), were generally characterized by positive POC anomalies, while anticyclones were generally associated with negative POC anomalies.

To further isolate the signature associated with each eddy[47] from other mesoscale features, we extracted the POC anomaly within 2-by-2 eddy radii from the eddy center (Supplementary Fig. 2c). To facilitate comparisons among eddies of various radii, the distance from the eddy center was normalized by the eddy radius on each 2-by-2 radii grid[20] (Supplementary Fig. 2c). Only eddies with at least 90% cloud-free pixel coverage for POC data within one eddy radius and 75% pixel coverage within two eddy radii (black box in Supplementary Fig. 2b) were used. To remove noisy, small-scale variability not related to the eddy, a 2D Gaussian function was fitted to the resulting POC anomaly field[47]. The fit is consistent with the average eddy shape that is well represented as Gaussian[1]. Last, the center of the Gaussian-fitted POC anomaly was shifted to align with the center of the eddy (Supplementary Fig. 2d). Repeating the analyses without using the Gaussian fit produced results that are qualitatively similar to those presented here (e.g., cyclonic eddies generated near the coast that propagated offshore are enriched in POC compared with those generated locally offshore). However, visual inspection of the POC anomaly fields indicated that large anomalies associated with other mesoscale features (e.g., upwelling front and filaments) are often observed around individual eddies, especially around the edges of the 2-by-2 eddy radii boxes, which influence the composites of the POC anomalies for each eddy amplitude bin (Fig. 3). Using the Gaussian fit allowed for the eddy signature to be isolated from the signature of these other mesoscale features.

In situ POC concentrations were measured in the CCS since 2006 as part of the CCS Long Term Ecological Research monitoring efforts[55]. Data availability is larger at the surface, decreasing significantly below 100-m depth. A depth range of 100 m for estimating organic carbon fluxes has been used in previous modeling studies[23]. In situ POC integrated from the surface to 100-m depth is correlated with surface concentrations ($r = 0.73$, $p < 0.01$; Supplementary Fig. 4), which allowed for the integrated POC content in the top 100 m to be estimated from satellite data following ref. [31]. Eddy-induced anomalies in the integrated POC content in the top 100 m (Supplementary Fig. 5) were extracted from each eddy following the procedure described above (Supplementary Fig. 2). Those anomalies were then used to calculate the offshore enrichment of POC (Gg year$^{-1}$) in the top 100 m by eddies (Fig. 6) as

$$\text{POC enrichment } (\text{Gg year}^{-1}) = \Delta(\text{POC}_{100}) \times \text{Eddy area} \times N, \quad (1)$$

where $\Delta(\text{POC}_{100})$ is the difference between the integrated POC content inside cyclones located offshore that were generated near the coast and those generated offshore for each amplitude bin (Supplementary Fig. 5), and $N$ is the number of eddies per year for each amplitude bin that are generated inshore of 300 km from the coast and propagate offshore. The POC enrichment was then summed for the multiple amplitude bins.

**Ocean model.** We use a previous regional implementation of the Regional Ocean Modeling System (ROMS[56]) to the CCS[57]. The model resolution is 4 km in the horizontal with 30 vertical terrain-following layers. Initial and boundary conditions for the regional model are obtained from a ROMS implementation to the entire North Pacific Ocean. The regional model is forced by surface wind stress from the SeaWinds scatterometer onboard NASA's Quick Scatterometer (QuikSCAT) satellite, and by heat and freshwater fluxes from NCEP North American Regional Reanalysis (NARR). Additional details of the model implementation are presented in ref. [57]. A passive tracer with unit concentration that does not sink was released uniformly throughout the entire water column inside four cyclonic eddies in different seasons (see Supplementary Fig. 3 for one example). These eddies in the model were generated near the coast and propagated offshore, and they had radii ranging from 55 to 85 km, which is consistent with characteristics of the satellite-detected eddies. The percentage of the tracer remaining inside each eddy in the top 400 m was then calculated as the eddy propagated offshore, which provided a measure of the fraction of the water effectively trapped by the eddy. The analysis revealed that $65 \pm 7\%$ of the water initially present in the interior of eddies generated near the coast remained trapped when those eddies reached over 300 km from the coast. This trapping efficiency was used to estimate the volume of water transported offshore by eddies in the CCS in the top 400 m as

$$\begin{aligned}\text{Volume transport (Sv)} \\ = (\text{eddy area} \times \text{trapping depth} \times N \times \text{trapping efficiency})/T,\end{aligned} \quad (2)$$

where $T$ is the number of seconds in 1 year. The volume transport was then summed for the multiple amplitude bins. Trapping efficiencies of 58 and 72% were also assumed to provide error bounds in the volume transport estimate. We note that the computation of POC enrichment in offshore regions (Eq. (1)) is not dependent on this model estimate of the trapping efficiency.

## Data availability

The data/reanalysis that support the findings of this study are publicly available online at http://wombat.coas.oregonstate.edu/eddies/index.html, https://oceandata.sci.gsfc.nasa.gov/SeaWiFS/, and https://doi.org/10.6073/pasta/20c05dd205be2225ecb32a5fede1c36c.

## Code availability

Model codes are available at www.myroms.org.

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

## Acknowledgements

We gratefully acknowledge support by NASA's Ocean Vector Winds Science Team (NNX14AM70G), Ocean Surface Topography Science Team (NNX13AD80G), Physical Oceanography (80NSSC18K0766) and Earth and Space Science Fellowship (80NSSC18K1342) programs, and by NSF (OPP-1643468).

## Author contributions

R.C. and P.M. conceived and designed the research; C.A. conducted the research; C.A. and R.C. wrote the paper; all authors commented on the paper.

## Competing interests

The authors declare no competing interests.
