## [Peer Review File · Nature Communications]

Reviewers' comments:

Reviewer #1 (Remarks to the Author):

This is a very nicely written paper that attempts to estimate the offshore flux of carbon by cyclonic eddies in the CCS. Overall I don't have any major issues with the paper and suggest it be published after addressing the following questions:

1) Your method of computing composite averages: I understand why you want to remove small-scale variability from the eddy signal, but I don't think that your method of doing so, by fitting a 2D Gaussian, is a great way to do it. I can imagine a scenario where the largest carbon concentration of carbon is located elsewhere than the eddy center, like along the edge. This could result from submesoscale variability or the fact that the smoothed altimetry products don't always get the eddy center in the correct place. In the event that the largest carbon signal was along the periphery of a cyclone, the 2D Gaussian would fit that as the max and then your methods would shift that to the center. This doesn't seem like a good idea as the stuff along the edge isn't necessarily trapped.

I would suggest you redo the analysis just using simple composite averages, and see what you get without the fancy 2D fitting. If it is much different, I think you will need to justify the use of the 2D fitting. If they are about the same, maybe include the comparison in the supplement.

2) The assumption that carbon is uniform over the upper 100 m is not valid when the mixed layer depth or the euphotic zone depth are less than 100 m. For example, my recent paper shows that cyclone can have winter time MLDs that are up to 20 m shallower than the climatological mean in the CCS. Anticyclones can have MLDs that are about 20 m deeper than climatology.

I would suggest that you put some error bound on your final carbon transport estimates by making these estimates for a range of depths, not just 100 m. It would be interesting to see what the estimates look like for anticyclones as well, seeing as 40 m more mixed layer depth would change the vertically integrated carbon content of anticyclones quite a bit compared to cyclones. This seems like a simple calculation that won't require much work and would help to highlight that we don't really know how eddies influence the vertical structure of phytoplankton communities, nor really how they modulate near surface stratification.

I am more than happy to speak with the authors about these comments and help in any way.

Here is a link to the paper I mention.

<https://agupubs.onlinelibrary.wiley.com/doi/full/10.1029/2018GL080006>

Dr. Peter Gaube

Reviewer #2 (Remarks to the Author):

This manuscript estimates the amount of particulate organic carbon transported offshore in eddies generated within the California Current System through analysis of satellite measurements of sea surface height and remote sensing reflectance. The authors identify that cyclonic eddies formed within 300 km of the coast have anomalously high POC content relative to anti-cyclonic eddies formed nearshore and relative to cyclonic eddies formed offshore and are responsible for the majority of offshore POC transport. The authors quantify the subsurface POC content through statistical relationships to surface values from in situ data and by counting eddies are able to quantify the offshore transport.

Overall I like the approach that the authors have attempted in order to measure something new. However, it is a challenging exercise, and I have questions about details regarding some of the estimates, outlined below. In addition, I think the authors came up short motivating the study, providing compelling justification for the significance of this particular work to a general audience. Overall, I found it hard to get sufficiently excited about the result. The combination of uncertainty in the estimates and insufficient motivation for the significance of their study lead me to not recommend publication in Nature as a general interest scientific journal. I recommend the authors resubmit to a more specific oceanography journal.

Major comments:

- 1) The analysis in this paper demonstrates that eddies generated nearshore influence ocean properties up to 1000 km from the coast, and the authors state that this is a meaningful and original conclusion and that prior estimates showed influence only out to 400 km (line 170). My general impression of the CCS is that coastal influences can be felt out much further than 400km. For example, Figure 12 from Marchesielo et al. (2003) demonstrates (from a modeling study) that SSH anomalies and the depth integrated EKE extends over 650 km from the coast. Plate 1 from Kelly et al. (1998) show Lagrangian drifters with eddying paths extending west of 135W, and Plate 2 shows non-negligible drifter EKE estimates out to 132W. Figure 4 from Sotka et al. (2004) shows 2-year drifter tracks out to 135W, with quite high density out to about 132W. I do not think people will be surprised by the result that eddy influences extend 1000 km.
- 2) The line fit in supplementary figure 5 shows such extensive scatter that I'm skeptical of the very small error bars in Figure 6c and d. The line fit has been used to estimate the subsurface POC concentration, required for the carbon transport estimates. Since they appear to be off by as much as 50%, it seems that the carbon transport uncertainty should be quite high as well, much higher than given. There is no uncertainty estimate on the estimate of 107 Gg C/year. It's possible that the uncertainty in subsurface information was propagated properly through the calculation, but it does not appear to be the case.
- 3) The reduction in carbon export from 107 Gg C/year to 12.4 Gg/year is not clear from the text. Based on the discussion around lines 76-80, nearshore cyclonic eddies account for about 13.4% of the total POC (and occupy about 10.4% of the total area) compared to 6.9% for offshore cyclonic eddies and 7% of the total area. Thus it appears safe to assume that nearshore cyclonic eddies have less than twice the POC content as offshore eddies. Given that number, it's hard to understand the ~90% reduction in offshore carbon anomaly transport (107 Gg/year -> 12.4 Gg/year). One might have expected only a factor of 2 reduction. More explanation of this surprising result is needed.
- 6) What is the uncertainty in the assumption of 400 m depth for the trapping scale? Does that vary in the cross-shore direction? Higher surface concentration nearshore could be associated with relatively shallow pycnocline depths, and a deepening offshore would result in lower surface concentrations as observed. Have the authors established that there is no trend in the pycnocline depth in the cyclonic eddies that would systematically bias their transport estimates? Also, is the uncertainty in the depth propagated through the calculations?
- 7) Is there any uncertainty in the calculated radius of the eddy, which will impact the volume as r^2 and is this error propagated through to the final estimates?
- 8) Onshore carbon transport by transient, but non-eddy, motion can not be accounted for by this method. Do you have any estimates for its scale? Perhaps it is small, but it may be a sizable fraction of the offshore POC transport.
- 9) The authors make the appropriate point (line 180) that the total carbon export is probably much higher than they can state through their estimate because dissolved concentrations can be

10-25 times higher than POC. This uncertainty calls into question the significance of the calculation overall and even the title of the paper, which implies it's a more comprehensive estimate.

Minor comments

No need to capitalize Eastern Boundary Current System in lines 39 and 47.

Reviewer #3 (Remarks to the Author):

Review of "Offshore transport of carbon in the California Current System by mesoscale eddies" by Amos et al.

> General comment

The paper by Amos et al. presents a study based on satellite data and focusing on the role of mesoscale eddies in the transport of organic carbon from the nearshore region of the California Current System to the open ocean. The authors provide a quantification of the export flux sustained by the eddies and discuss the lateral isolation of mesoscale eddies with the use of model data and of a modelled passive tracer experiment.

The topic on which the authors focus is highly relevant and interesting. There are currently only a limited number of studies focusing on the role of mesoscale activity in the lateral transport of the organic transport, and therefore the study has potential to provide valuable information on the subject.

However, I have several fundamental concerns as regard to the methods and conclusions drawn by the authors on the basis of their analysis of the organic carbon concentration in the eddies. Also, the discussion of the results does not include an in depth analysis of the potential limitations of the adopted methods, both regarding the use of satellite data and regarding the parallel with the modelling study. A few choices in terms of analysis methods are not well justified.

Furthermore, the authors forget to refer to some fundamental publications on the topic which already focused on the subject of eddy isolation and eddy organic carbon transport in upwelling regions, therefore lacking a thorough discussion of their results in the context of the present findings.

Unfortunately, I don't think that this study is yet ready for publication.

> Major comments

1. Biogeochemical fluxes are overlooked

The authors discuss the lateral eddy transport from the coast on distances of several hundreds of km. A typical eddy drifts at a speed of a few cm/s, meaning it takes months to cover a distance of 500 km, as confirmed by the authors at lines 84-85. Despite this time lag and the biological processes affecting POC concentrations, the authors use high particulate organic carbon (POC) concentrations as an indicator of the presence of coastal water in the eddies (page 4, lines 1-3), and state that the high organic carbon concentration of eddies found hundreds of km offshore proves that this organic carbon was trapped and transported from the coast to the open waters (page 4 lines 85-88). The authors forget to discuss thoroughly the substantial biogeochemical transformations that the organic carbon goes through in the course of a few months. There is only a quick discussion of production in the eddies at lines 152-157, where the authors compare POC concentrations of nearshore and offshore generated eddies and use their difference to quantify the lateral transport of coastal organic carbon.

This comparison overlooks the many differences between these two populations of structures found offshore, one younger (offshore generated eddies) and one older (nearshore generated eddies that drifted offshore in time). It also overlooks the potential differences in biogeochemical properties such as POC, nutrient concentrations and rates of production due to the different regions of formation, and finally overlooks the simple fact that the coastal carbon in older eddies may have sunk while they were drifting offshore. I therefore think that this comparison used to

calculate the portion of coastal carbon that reaches the offshore region is not solid.

It would have actually been interesting, if not necessary, to know the mean properties and number of the onshore and offshore generated eddies that were analysed, and to see the equations used to calculate this net transport, as at the present moment the mathematical details of this calculation are not clear to me.

As regard to the biogeochemical transformations of POC in drifting eddies, there are different pathways through which the organic carbon trapped at the coast may not reach the offshore waters, while the eddy is nevertheless showing a high POC signature. I strongly suggest that the authors better consider and discuss these possibilities and their implications for their study. Recycling: the carbon trapped in the coastal region may have been locally remineralized and recycled into new carbon. Vertical export and new production: the organic carbon trapped at the coast may have sunk, while the organic carbon visible offshore may have been produced on the way to the open ocean through the utilization of the nutrients trapped at formation. The paper lacks a satisfactory discussion of these biogeochemical transformation and the conclusions may be affected by this shortcoming.

In order to overcome these limitations, the authors could use a different approach to the problem. The authors could change perspective and focus on the quantification of the eddy lateral transport of POC, compared to the total lateral transport (eddy and non eddy) of the same tracer in the region, without assuming that the POC found in the eddies was produced in the nearshore. This would still be a highly relevant quantification of the eddy lateral fluxes and of their potential contribution to the organic carbon cycle in the region. It is nearly impossible to determine where the carbon found in the eddies offshore was really produced near the coast, given these data. Such a change of focus would allow the authors to provide a much more solid quantification of the role of the eddies in the organic carbon lateral relocation, without the need of uncertain assumptions.

2. Concern regarding the use of satellite data

The method used by the authors for the quantification of the organic carbon content of the eddies is based on the use of satellite POC data combined with the global eddy database made available online by Chelton et al. The idea is straightforward, and in line with several other previous studies of the eddy properties. However, a few recent studies have raised concern regarding the use of satellite data for the analysis of mesoscale activity. In particular, the paper by Amores et al. (2018) titled "Up to what extent can we characterize ocean eddies using present-day gridded altimetric products?" highlights an important underestimation in the number of eddies and most of all significant distortions in the identified eddies which result in the identification of fake large eddies. I strongly suggest to add the discussion of the uncertainties connected to such potential errors in the paper and possibly provide a rough estimate of the error on the flux quantification.

3. Concern regarding the model-based quantification of trapping in eddies

Part of the analysis is built on a quantification of the trapping and transport capacity of cyclonic eddies obtained through the use of some rather high resolution modelling study. This quantification of trapping in the eddies is made by releasing a passive tracer inside of the structures and calculating the portion of this tracer that is retained by the structure in time. However, I have doubts regarding the applicability of these results in the present study. First of all, there is not much information about the size of the sample of modelled cyclonic eddies that was studied. Also, with such a high resolution grid, possibly resolving small eddies of the size of a few tens of km, I wonder if the modelled eddy population compares to the eddy population identified in satellite data, which typically includes only large eddies. How many cyclonic eddies were studied? What is the size distribution of these eddies, their non-linearity parameter, and how does it compare to that of the eddies identified from the satellite?

Another, maybe more serious, concern regards the properties of the passive tracer released in these modelled eddies, which is not well illustrated in the paper. Does this passive tracer sink? This may be a very critical issue in terms of applicability of the model results, as POC (differently from dissolved tracers) can sink out of the eddies at speeds between 1 m/day to a above 100 m/day, therefore escaping from the first 400 m of depth pretty quickly. Also, the authors study the tracer that remains trapped in the first 400 m of depth, but they don't specify is the tracer was

released only in the surface of the eddy or also at depth. I believe that more details regarding the experiments must be provided in order to assess whether if the resulting percentage of retained tracer can be used for the study of a sinking tracer such as POC, which is largely trapped and produced also below the surface.

4. Eddy finding & trapping algorithm choice and performance

I would have appreciated to find some essential information (at least in the methods section) on the type of algorithm originally used by Chelton et al. for finding & tracking the eddies, as the entire study is based on the use of Chelton et al.'s eddy database. The authors could summarize some essential information about this eddy finding & tracking algorithm and of its performance, as their results may be sensitive to the algorithm's ability to correctly identify and track the eddies. For example, the distinction between coastally- and offshore-generated eddies is sensitive to the presence of broken tracks that seem to originate offshore but are actually parts of longer tracks generated near the coast. This is generally carefully checked by observing an animation of the tracks on SSH. Was this checked by the authors?

In general, eddies are generated by flow instabilities (coastal shear, currents hitting obstacles...) or wind stress curl injection, therefore I would expect less eddies to be generated away from the coast. Of these, many of them may be detaching from long coastal filaments and therefore could still trap coastal water (filaments often exceed 300 km of offshore length, especially in the California Upwelling). How many tracks are generated offshore of 300 km, and what mechanisms of generation are responsible for their formation?

Also, the authors select only the highly non-linear eddies from the entire eddy database (page 11, lines 236-240). However, since the authors use eddy tracks, it is not clear how are the slowly rotating eddies removed from the downloaded tracks, as the same eddy may speed up and slow down along its path. Do the authors use only portions of the tracks in which the eddy is sometime not highly nonlinear? How many tracks are studied in total? More information on this part of the methods may be useful to the reader.

5. Choice of the nearshore/offshore domains

The authors identify the 300 km offshore line as the boundary between "nearshore" and "offshore" waters for the track generation. However, it is not clear to me why 300 km offshore should represent an important boundary. Is there a physical or biogeochemical divide between the region closer and further than 300 km from the coast? At page 11, lines 242-244 the authors quickly state that their results would not change if the boundary was instead placed at 350 km offshore. However, there is no motivation to support this other potential choice of offshore distance either. The authors should justify their choices better. Why are the first 300 km from the coast the "nearshore" region? Why not 200 km offshore or any other range of distances?

6. It is not clear what the authors mean by "export"

The authors focus on the lateral transport of organic carbon by the eddies from the coast to the open waters, and mention the "total export" by the eddies (e.g., abstract, line 33). However, the word export is generally used to refer to the vertical relocation of the organic carbon from the euphotic layer (or mixed layer) to below. Do the author always refer to the lateral transport only? If so, I'd suggest to always add the adjective "lateral" as in "lateral export", or to use expressions such as "lateral transport" or "lateral redistribution". Otherwise, I'd suggest to clarify better when the study refers to lateral transport, and when to vertical export.

Also, the authors talk about export from 0-300km offshore to 300km-900km offshore. It is not clear to me what a lateral export into an offshore box represents mathematically, as the lateral export is essentially a flux, and, as such, it is calculated as "concentration*velocity*surface" (and not *volume). I would rather talk about lateral export through the 300 km boundary, and not into the 300km-900km box. I am not sure if I understand the meaning of this flux, unless it is a vertical export. How is this export calculated? Could the author refer explicitly to the used mathematical method or equations?

7. Eddies or cyclonic eddies?

The authors essentially talk about cyclonic eddies rather than all the eddies, as one would expect from the paper's title. In fact, they even move the anticyclonic eddy plots into the supplementary material. Even though the anticyclonic eddies do not have such an important role in the transport of POC, I'd suggest not to "hide" those plots, or, alternatively, I'd change the paper title and abstract explicitly focusing on the cyclonic eddy population only.

7. Missing discussion of recent relevant literature

The paper discusses the interesting question of the role of eddies in the organic carbon transport, however the authors do not refer to some recent publications on the topic, and therefore miss the opportunity to compare their results to those of these previous studies.

On the topic of the eddy lateral isolation and eddy vertical structure, the authors could compare their modelling results to observational studies of the eddy structure and lateral isolation, such as:

- Amores et al. 2016 "Coherent mesoscale eddies in the North Atlantic subtropical gyre: 3-D structure and transport with application to the salinity maximum"

- Pegliasco et al. 2015 "Main eddy vertical structures observed in the four major Eastern Boundary Upwelling Systems"

A similar (but model-based) study to the present one was presented by:

- Lovecchio et al. in 2018 "Mesoscale contribution to the long-range offshore transport of organic carbon from the Canary Upwelling System to the open North Atlantic".

This study quantifies the eddy and filament lateral transport of organic carbon from the nearshore region of the Canary Upwelling system. How do your results compare to this study?

Further relevant studies are:

- Capet et al. 2013 "Eddies in Eastern Boundary Subtropical Upwelling Systems"

- Chaigneau et al. 2009 "Eddy activity in the four major upwelling systems from satellite altimetry"

> Detailed comments

Page 2, line 43: Please, define what you mean by nonlinear (e.g., rotational speed/drift speed >1?). Also, I'd say "of these eddies".

Page 4, line 75: Please, state explicitly that you are using Chelton's database of already identified and tracked eddies. Also, I suggest to add more details about how these eddies were found and tracked by Chelton, in order to provide info on the used algorithm, its strengths and its limitations.

Page 4, lines 77-80: According to Chaigneau et al. 2009 (<https://doi.org/10.1016/j.pocean.2009.07.012>), eddies occupy about 24%-30% of the surface of upwelling regions, with the highest value found in the California Upwelling System. Why is the eddy surface fraction so low in the current study?

Page 4, lines 85-88: This sentence is too speculative and not well grounded. In 3-6 months POC may have gone through intense bgc transformations, it is not likely that the POC you find in an eddy months after its formation corresponds to the POC trapped at formation.

Page 5, lines 95-98: Please, do not put anticyclonic eddy plots in the supplementary, unless you focus your paper explicitly on cyclones. Also, differences in the water trapped by cyclones and anticyclones are not restricted to POC, but also include nutrients and living beings. I suggest the authors to keep these differences in mind, and consequently discuss the time evolution of the tracers in the eddies.

Page 5, line 103 and Methods lines 206-221: POC anomalies are calculated as differences between the satellite-derived weekly mean POC field and the same field smoothed in space with a low pass filter. First off, there is no information about what are the properties of such a filter, how many

points are averaged together in space, therefore results could be strongly influenced by the choice of the filter's properties. Second I find this method improper, as the POC value in each grid point is still influenced by the POC contained by a potential eddy in that point even after smoothing. This is not a valid reference field from which to calculate anomalies. With 13 years of satellite data available, the authors should use a valid reference mean as an annual mean POC field, or even better a monthly mean POC field interpolated weekly. I believe that the anomalies calculated with the use of a low pass filtered field as a reference field are not significant measures of the POC concentration in the eddies. Also, I don't believe that applying a gaussian filter to the calculated eddy anomalies is necessary, nor it is meaningful to regrid the mean eddy anomalies onto a higher resolution normalized grid, as no new information is generated through regridding of low resolution data.

Page 5, line 104: Why at 300 km from the coast and not 100 km or 200 km or any other distance? How do you define this boundary and what is its relevance? (same for Methods, lines 242-244)

Page 5, line 110: Analogously, why 300 – 600 km offshore?

Page 5, lines 110-111: The high POC concentration found offshore in eddies generated near the coast does not indicate that the carbon trapped at formation is still in the eddies many months later. That carbon may have completely sunk, and have been substituted by newly generated carbon through the use of trapped nutrients.

Page 5, line 114: Why now 900-1200km offshore, and what happens in the 600-900 km region that is never mentioned? I suggest to clearly define these boundaries before discussing the results or in the methods and to justify their choice, as above. Also, it would be useful to associate these distances to the time taken by the eddies to reach them, for example on average in the entire eddy track population.

Page 6, lines 118-119: This is too speculative. POC could also be remineralized, as well as it may sink. It is simply not possible to assess the biogeochemical transformations from satellite data. I suggest either to discuss all the possibilities and their consequences for the results, or not to attempt to pick one single explanation for the POC tendency. Also, the fact that POC decreases in time is expected.

Page 6, lines 128-129: This sentence is unclear, please rephrase it.

Page 6, line 134: Too speculative. Also, how many eddies are really generated offshore of 300 km from the coast and by which mechanisms? Are they not just pinching from long filaments? (which would indicate that they also trap some coastal water) Are there islands that perturb the flow? What is the seasonality of this offshore eddy generation? Again, what is the significance of distinguishing between eddies generated closer and farther than 300 km from the coast?

Page 6, line 138: 6 cyclones per year sounds a bit low. Is this because you are only looking at a subset of the entire eddy tracks population?

Page 7, line 140: In your modelling study, did you restrict your analysis to large eddies comparable to those found in satellite data?

Page 7, lines 146-150: Why do you need in situ POC when you have the vertically integrated POC from satellite data already available, which would allow you to make explicitly the same flux calculation? How many in situ measurements do you use per each eddy?

Page 7, lines 152-157: This approach is not solid, as discussed above in the major comments

Page 8, lines 169-170: As the authors state, filaments extend easily up to 400 km offshore. Eddies

often detach from these filaments, trapping coastal upwelled water that is transported far offshore by the filaments. Therefore, again, distinguishing between eddies generated at offshore distances smaller and larger than 300 km offshore does not necessarily separate eddies that are influenced by upwelled waters and eddies that are not.

Page 9, lines 184-186: The high POC concentration and potential remineralization of the eddy POC could also lead to degassing of CO₂ from the eddies. The discussion is just too speculative and sounds out of context.

Reviewer #1 (Remarks to the Author):

This is a very nicely written paper that attempts to estimate the offshore flux of carbon by cyclonic eddies in the CCS. Overall I don't have any major issues with the paper and suggest it be published after addressing the following questions:

We are glad the reviewer enjoyed our manuscript, and we thank him for the comments and suggestions. A detailed description of how those were addressed is presented below.

1) Your method of computing composite averages: I understand why you want to remove small-scale variability from the eddy signal, but I don't think that your method of doing so, by fitting a 2D Gaussian, is a great way to do it. I can imagine a scenario where the largest carbon concentration of carbon is located elsewhere than the eddy center, like along the edge. This could result from submesoscale variability or the fact that the smoothed altometry products don't always get the eddy center in the correct place. In the event that the largest carbon signal was along the periphery of a cyclone, the 2D Gaussian would fit that as the max and then your methods would shift that to the center. This doesn't seem like a good idea as the stuff along the edge isn't necessarily trapped.

I would suggest you redo the analysis just using simple composite averages, and see what you get without the fancy 2D fitting. If it is much different, I think you will need to justify the use of the 2D fitting. If they are about the same, maybe include the comparison in the supplement.

We appreciate the reviewer's concern with using the 2D Gaussian fitting. Following the reviewer's suggestion, we redid the analyses using the composite averages of the POC anomalies without using the 2D Gaussian (Figure A.1). The POC content averaged across the box shows a similar pattern to the averages computed using the 2D Gaussian fit (Figure 3 in the manuscript). Cyclonic eddies generated inshore that are still located inshore have the highest POC anomalies as before (Figure A.1a). Cyclonic eddies generated inshore that are located offshore have higher concentrations of POC on average than eddies generated locally offshore (Figure A.1b-d), which is consistent with the results obtained when the 2D Gaussian fit is applied (Figure 3d in the manuscript). Note that the magnitude of the POC anomalies is different, however, because in Figure A.1 they were computed over the entire 2 x 2 radii box (instead of 1 eddy radius), since the POC anomalies and the eddy centroid are not always coincident (note also that this can even lead to negative values, if a large negative anomaly is located inside the 2 x 2 box, as in Figure A.1c).

The composites of the original POC anomalies (without the Gaussian fit) show patches of POC along the edges of the 2 by 2 box that are not related to the eddies (upper right corner of Figure A.1a-c). Without the 2D Gaussian fit, it is very difficult to remove the influence of non-eddy mesoscale features. In the CCS, there is strong variability in POC associated with several mesoscale processes, such as eddies, the upwelling front, filaments, etc. There is overlap in the spatial scales of these processes – both eddies and the upwelling front scale with the internal Rossby radius (Kundu et al., 1975; Kamenkovich et al., 1986; Pedlosky, 1987; Cushman-Roisin and Tang, 1990). If we use a spatial filter that removes the influence of the upwelling front, it will also remove a lot of the eddy signature. In order to keep the eddy signature, then features

associated with the upwelling front will still be present, at least partially, which often results in large POC anomalies along the edges of the 2 by 2 box as seen in Figure A.1a. Therefore, we find it necessary to apply the 2D Gaussian fitting to isolate the POC signature associated with eddies. It is reassuring that, regardless of the method used, cyclones generated inshore that propagate offshore are found to be enriched in POC compared to those generated offshore (Figure 3 and A.1).

Supplementary Figure 2c in the manuscript also shows a good example of why we need to isolate the eddy signature using the Gaussian fit. The band of high POC values near the northeastern corner of the box is not related to the eddy, but these values would influence the composite averaging of the original POC without the Gaussian fitting. We note also that in this case the eddy centroid based on the altimetry and the POC data are nicely aligned, but this is not always the case. The eddy centroid was computed by Schlax and Chelton (2016) as

$$x_c \equiv \frac{\sum_{(i,j)} x(i)h(i,j)}{\sum_{(i,j)} h(i,j)}$$

$$y_c \equiv \frac{\sum_{(i,j)} y(j)h(i,j)}{\sum_{(i,j)} h(i,j)}$$

where h is the 2D field of SSH from altimetry, the index i corresponds to a specific longitude $x(i)$ and the index j corresponds to a specific latitude $y(j)$. In the example in Figure A.2a, the eddy centroid is not coincident with the region of maximum sea level anomaly, but there is a clear POC signature associated with the region of maximum SLA (Figure A.2a). When the POC anomaly is extracted within 2 by 2 eddy radius, the POC signature is not coincident with the eddy centroid (Figure A.2b). Note that high POC anomaly associated with a coastal filament is also present in the northeastern corner of the box spanning 2 by 2 radii. Attributing this entire POC anomaly (2 x 2 box) to the eddy would be inaccurate. Considering only the POC anomaly within 1 eddy radius (i.e., inside black circle in Figure A.2b) would also be inaccurate, since part of the POC anomaly associated with the eddy on the left side of the box would be neglected. By applying the 2D Gaussian fit we can isolate the signature of the POC associated with the eddy (Figure A.2c), however, and shift this to the center of the box. We acknowledge/agree that the 2D Gaussian fitting introduces some smoothing of the POC signatures in the analyses and that we are losing the smaller scale variability (although we note that the resolution of the ocean color data – 9 km – precludes submesoscale variability to be captured). We have modified the text to emphasize this.

With regard to the reviewer’s concern that in cases where the largest POC signal is found along the periphery of a cyclone, the 2D Gaussian would fit that as the max and then the methods would shift that to the center, we tested the 2D Gaussian fit in this scenario to see the extent that this is true. We simulated an eddy with a higher concentration of POC along the eddy periphery, then applied the 2D Gaussian fit (Figure A.3). Since that scenario cannot be well represented by a Gaussian, the concentration after applying the Gaussian fit is significantly lower than the original concentration along the periphery (Figure A.3b). So even in cases where the Gaussian fits to a large POC signature along the eddy periphery and shifts this to the center, this does not

affect the composite averages substantially since the POC anomaly based on the Gaussian fit is small in those cases (compare the colorbar in Figure A.3b with Figure A.2c, for example).

Figure A.1 – (a-c) Composites of original POC anomalies (*without* applying 2D Gaussian fit) within 2 by 2 eddy radii for cyclonic eddies with amplitudes 9-11 cm. (d) Mean POC anomaly (mg m⁻³) and standard error calculated within 2 by 2 eddy radius grouped by eddy amplitude.

Note that this is different than in Figure 3d in the manuscript, where POC anomaly was computed within 1 eddy radius. See text above for details.

Figure A.2 – (a) POC anomaly (mg m⁻³) and sea level anomaly contours at 1 cm intervals on 2 December 2012. Solid contours are negative. Black box marks the region within 2 by 2 eddy radii from the eddy centroid (grey circle) as identified by altimetry (Chelton et al., 2011). (b) POC anomaly extracted within the black box in panel a. Note that high values on the upper right corner of box are not directly related to the eddy. (c) POC anomaly as determined by Gaussian fit.

Figure A.3 – (a) Simulation of eddy with largest concentration of POC located along the eddy periphery. Inner white circle marks 1 eddy radius. Outer white circle marks 1.15 eddy radius. (b) 2D Gaussian fit applied to the eddy simulated in panel a. White circle marks 1 eddy radius.

2) The assumption that carbon is uniform over the upper 100 m is not valid when the mixed layer depth or the euphotic zone depth are less than 100 m. For example, my recent paper shows that cyclone can have winter time MLDs that are up to 20 m shallower than the climatological mean in the CCS. Anticyclones can have MLDs that are about 20 m deeper than climatology.

I would suggest that you put some error bound on your final carbon transport estimates by making these estimates for a range of depths, not just 100 m. It would be interesting to see what the estimates look like for anticyclones as well, seeing as 40 m more mixed layer depth would change the vertically integrated carbon content of anticyclones quite a bit compared to cyclones. This seems like a simple calculation that won't require much work and would help to highlight that we don't really know how eddies influence the vertical structure of phytoplankton communities, nor really how they modulate near surface stratification.

Following the reviewer's suggestion, we have calculated the POC enrichment offshore due to eddy lateral transport for a range of depths. For that, we first estimated the relationship between surface POC and integrated POC for different depth ranges (e.g., top 20 m; top 50 m), analogously to the calculation done for the top 100 m shown in Supplementary Figure 4. The estimated POC enrichment due to cyclonic eddy transport for a surface layer 20, 50 or 100 m thick is 5.4 ± 2.9 , 12.2 ± 6.1 or 20.9 ± 11 Gg C/year, respectively. We have added that information to the text, thanks for the suggestion. We note that the correlation between surface POC and the integrated POC for different depth ranges is approximately depth independent, except for a small increase in the top 20 m (Figure A.4).

With regard to anticyclonic eddies, we re-did the analyses for 20 and 50 m too. We found in our analyses that anticyclones generated near the coast are generally not associated with large positive POC anomalies and are not trapping as much of the recently upwelled water during formation like the cyclones. It is possible that anticyclones can trap upwelled coastal waters in some cases, but that is not happening in most cases. Therefore, when we make the composites of the POC anomalies associated with anticyclones, the signature of POC transport offshore is not apparent. That is consistent with results shown in Figure 2 in the manuscript, which indicates that in contrast to cyclonic eddies, there is not a relative enrichment of POC inside anticyclones compared to the area occupied by those anticyclones (compare Figure 2c and 2d).

Figure A.4 – Correlation coefficient between surface POC and integrated POC for different depth ranges (top 10 m, top 20 m, top 30 m, etc.).

I am more than happy to speak with the authors about these comments and help in any way.

We thank the reviewer again for the helpful suggestions. As a final note, our final estimate of POC enrichment due to cyclonic eddy lateral transport has been changed to 20.9 ± 11 Gg C/year. The difference occurs because in the original version of the manuscript, the distance of 300 km from the coast had been incorrectly calculated by simply computing that distance in the zonal direction. In the revised manuscript, the distance of 300 km is calculated in the direction perpendicular to the coastline. Although the difference between the two methods is small in most regions, it is actually quite large near Point Conception (Supplementary Figure 1). As a result, many of the eddies generated within 300 km from the coast in that region had been mistakenly assigned as being generated in the offshore region in the original manuscript. That artificially increased the POC content of eddies generated offshore, resulting in a smaller estimate of POC enrichment due to eddy transport (according to Eq. 1 in the manuscript). This has been fixed in this revised version. We apologize for the confusion.

Here is a link to the paper I mention.

<https://agupubs.onlinelibrary.wiley.com/doi/full/10.1029/2018GL080006>

Dr. Peter Gaube

Reviewer #2 (Remarks to the Author):

This manuscript estimates the amount of particulate organic carbon transported offshore in eddies generated within the California Current System through analysis of satellite measurements of sea surface height and remote sensing reflectance. The authors identify that cyclonic eddies formed within 300 km of the coast have anomalously high POC content relative to anti-cyclonic eddies formed nearshore and relative to cyclonic eddies formed offshore and are responsible for the majority of offshore POC transport. The authors quantify the subsurface POC content through statistical relationships to surface values from in situ data and by counting eddies are able to quantify the offshore transport.

Overall I like the approach that the authors have attempted in order to measure something new. However, it is a challenging exercise, and I have questions about details regarding some of the estimates, outlined below. In addition, I think the authors came up short motivating the study, providing compelling justification for the significance of this particular work to a general audience. Overall, I found it hard to get sufficiently excited about the result. The combination of uncertainty in the estimates and insufficient motivation for the significance of their study lead me to not recommend publication in Nature as a general interest scientific journal. I recommend the authors resubmit to a more specific oceanography journal.

We thank the reviewer for his/her comments and suggestions which were very useful in revising the manuscript and clarifying aspects that were unclear before. We have added text to the manuscript to further explain the motivation for this study. One of the key points is that previous studies of eddies trapping and transporting water relied mostly on models or on theoretical arguments (or in observations in isolated events). To our knowledge, this is the first observational evidence of this process over multiple events demonstrating that it is significant enough to alter the composite distribution of properties in the CCS. We have clarified this in the manuscript, and we thank the reviewer for pointing out that this was not clear.

Major comments:

1) The analysis in this paper demonstrates that eddies generated nearshore influence ocean properties up to 1000 km from the coast, and the authors state that this is a meaningful and original conclusion and that prior estimates showed influence only out to 400 km (line 170). My general impression of the CCS is that coastal influences can be felt out much further than 400km. For example, Figure 12 from Marchesiello et al. (2003) demonstrates (from a modeling study) that SSH anomalies and the depth integrated EKE extends over 650 km from the coast. Plate 1 from Kelly et al. (1998) show Lagrangian drifters with eddying paths extending west of 135W, and Plate 2 shows non-negligible drifter EKE estimates out to 132W. Figure 4 from Sotka et al. (2004) shows 2-year drifter tracks out to 135W, with quite high density out to about 132W. I do not think people will be surprised by the result that eddy influences extend 1000 km.

The comment about ~400 km refers to the offshore deflection of the upwelling jet and filaments, which are capable of advecting coastal water offshore. We agree that eddies can propagate much farther than that, but that does not mean that they are transporting coastal water offshore – that only happens if the eddies are indeed trapping water and transporting it as they propagate.

It is true that increased altimeter EKE in Plate 2a of Kelly et al. (1998) extends to about 130-132°W, but the highest values occur inshore of 128°W (~350 km from the coast). For the drifter EKE (Plate 2b), there is enhancement to 132°W (650 km from the coast), but the largest values are found inshore of 128°W. It is also not clear from these studies that nutrients or carbon are enhanced in the interior of these mesoscale features. The altimetry data (left panel of Plate 2) does not necessarily show trapping, just that a feature with increased EKE is located at a given region (e.g., it could be due to a linear eddy that is not trapping materials). Our key result is not that eddies propagate to 1000 km from the coast, which we agree has been demonstrated many times in the literature, but that cyclonic eddies can trap materials and transport them offshore, maintaining a detectable signature to about 1000 km from the coast. We have revised the text to clarify this important point. We thank the reviewer for pointing out that this was not clear.

2) The line fit in supplementary figure 5 shows such extensive scatter that I'm skeptical of the very small error bars in Figure 6c and d. The line fit has been used to estimate the subsurface POC concentration, required for the carbon transport estimates. Since they appear to be off by as much as 50%, it seems that the carbon transport uncertainty should be quite high as well, much higher than given. There is no uncertainty estimate on the estimate of 107 Gg C/year. It's possible that the uncertainty in subsurface information was propagated properly through the calculation, but it does not appear to be the case.

We thank the reviewer for questioning the uncertainties in the calculations. We added the uncertainty in the slope and y-intercept from the linear fit to Supplementary Figure 4. The uncertainty in the linear fit was not originally propagated through the carbon transport calculation. The initial calculation included uncertainties in the integrated POC anomalies associated with the eddies in each amplitude bin. Our calculation of the POC enrichment in offshore regions now includes the uncertainties we had considered previously, as well as uncertainties in the linear fit, eddy radius, and the number of eddies per year.

3) The reduction in carbon export from 107 Gg C/year to 12.4 Gg/year is not clear from the text. Based on the discussion around lines 76-80, nearshore cyclonic eddies account for about 13.4% of the total POC (and occupy about 10.4% of the total area) compared to 6.9% for offshore cyclonic eddies and 7% of the total area. Thus it appears safe to assume that nearshore cyclonic eddies have less than twice the POC content as offshore eddies. Given that number, it's hard to understand the ~90% reduction in offshore carbon anomaly transport (107 Gg/year → 12.4 Gg/year). One might have expected only a factor of 2 reduction. More explanation of this surprising result is needed.

We appreciate the reviewer's confusion with these transport estimates. In the original version of the manuscript, we presented 2 estimates, 107 and 12.4* Gg C/year. The smaller estimate was obtained by comparing the integrated POC content between eddies that are generated locally offshore and those that are generated near the coast and then propagated offshore. This gives us the amount of additional POC that is added to the offshore region due to the trapping and transporting of upwelled coastal water by eddies as they propagate offshore.

The other estimate, of 107 Gg C/year, considered only the amount of POC inside the eddies generated near the coast that propagated offshore. We agree that this number is meaningless – an

enrichment of 107 Gg/year would only occur if eddies generated locally offshore had zero POC content, which we know is not true. Because this estimate is meaningless, we removed it from the manuscript to avoid confusion. We thank the reviewer for pointing out that this was confusing. We have modified the text to clarify this issue.

The other number referred to by the reviewer, the percentage of the total POC inside cyclones (e.g., 13.4%), is dependent on the amount of POC *both inside and outside* of the eddies in the domain. For example, an increase in POC outside of the eddies, which could be related to a filament for example, will result in a decrease in the percentage of POC that is found inside the eddies (since the sum of both quantities must equal 100%). Those percentages are not used to compute POC enrichment/transport, but rather to show that cyclones formed inshore that propagate offshore are different in terms of POC content compared to cyclones formed offshore (compare Figure 2a and 2c).

*Note: The estimate of POC enrichment offshore due to cyclones has been updated to 20.9 ± 11 Gg C/year. The difference occurs because in the original version of the manuscript, the distance of 300 km from the coast had been incorrectly calculated by simply computing that distance in the zonal direction. In the revised manuscript, the distance of 300 km is calculated in the direction perpendicular to the coastline. Although the difference between the two methods is small in most regions, it is actually quite large near Point Conception (Supplementary Figure 1). As a result, many of the eddies generated within 300 km from the coast in that region had been mistakenly assigned as being generated in the offshore region in the original manuscript. That artificially increased the POC content of eddies generated offshore, resulting in a smaller estimate of POC enrichment due to eddy transport (according to Eq. 1 in the manuscript). This has been fixed in this revised version. We apologize for the confusion.

6) What is the uncertainty in the assumption of 400 m depth for the trapping scale? Does that vary in the cross-shore direction? Higher surface concentration nearshore could be associated with relatively shallow pycnocline depths, and a deepening offshore would result in lower surface concentrations as observed. Have the authors established that there is no trend in the pycnocline depth in the cyclonic eddies that would systematically bias their transport estimates? Also, is the uncertainty in the depth propagated through the calculations?

We agree with the reviewer that the trapping depth may vary in the cross-shore direction, but we emphasize that we are *not* using the trapping depth in the calculations of POC enrichment offshore (see Eq. 1 in manuscript). Since we can only compute the integrated POC content in the top 100 m, our estimates of POC enrichment in the offshore region due to eddy transport is restricted to that 100 m limit. Thus, as long as the trapping depth for the eddies is larger than 100 m, the calculation is self-consistent. This is also consistent with Lovecchio et al. (2018), who also used 100 m in their modeling study.

We only use the trapping depth of 400 m when computing the volume of water transported by eddies (Eq. 2). Kurian et al. (2011) state that cyclonic eddy effects are noticeable down to 400 m and their Figure 16 shows that vorticity (ζ/f ; top panel) is largest in the top 400 m for cyclones. This value from Kurian et al. (2011) is consistent with our modeling results. Using 400 m as the trapping depth, we get a volume transport of ~ 1 Sv, which is on the same order as the

transport associated with the upwelling jet. If we had used a trapping depth of 300 m or 500 m, we would get ~ 0.8 Sv and ~ 1.3 Sv, respectively, which is still on the same order as the jet. Our goal here was to demonstrate that the volume of water transport by eddies is comparable to the volume transport due to other processes.

We have modified the text to make these points clearer. We thank the reviewer for pointing out that this was confusing.

7) Is there any uncertainty in the calculated radius of the eddy, which will impact the volume as r^2 and is this error propagated through to the final estimates?

We thank the reviewer for questioning this. Uncertainty in eddy radius (Figure B.1) was not originally propagated in the calculations for the volume and POC enrichment/transport. We have recalculated the transports and the errors to include that.

Figure B.1 – Average eddy radius (km) and standard error as a function of eddy amplitude. These averages and uncertainties were used to estimate the eddy area/volume that was used in the volume transport and POC enrichment calculations.

8) Onshore carbon transport by transient, but non-eddy, motion can not be accounted for by this method. Do you have any estimates for its scale? Perhaps it is small, but it may be a sizable fraction of the offshore POC transport.

We do not have estimates for non-eddy motions but filaments and small processes are also important features likely contributing to offshore transport. Current satellite products are too low resolution to resolve submesoscale variability. Filaments can often be seen with satellite products; however we do not have a method to compute lateral carbon transport for these features. This can be done with high-resolution three-dimensional models (e.g. Nagai et al., 2015; Lovecchio et al., 2018), but not with satellite data alone.

9) The authors make the appropriate point (line 180) that the total carbon export is probably much higher than they can state through their estimate because dissolved concentrations can be 10-25 times higher than POC. This uncertainty calls into question the significance of the calculation overall and even the title of the paper, which implies it's a more comprehensive estimate.

We agree that we are not calculating the lateral total organic carbon transport. We are using POC as a tracer of the recently upwelled coastal water that contains nutrients, POC, DOC, and other

properties. We are focusing on the particulate fraction because that is what can be readily estimated by satellites. We discuss that the dissolved fraction is larger, and it is plausibly also being transported offshore by cyclones. The same is true for other quantities that are not measurable by satellites (e.g., nutrients). We have modified the text to specify that the calculations are of POC, not total carbon, and modified the title to address this concern. Rather than provide an estimate of total organic carbon transport, an unachievable goal at the moment, our main goal is to demonstrate observationally that trapping of coastal water by nonlinear cyclones and subsequent transport has a significant effect on the POC distribution in offshore regions of the CCS.

Minor comments

No need to capicalize Eastern Boundary Current System in lines 39 and 47.

We made this change in the text.

We thank the reviewer again for the thoughtful comments and suggestions.

Reviewer #3 (Remarks to the Author):

Review of "Offshore transport of carbon in the California Current System by mesoscale eddies" by Amos et al.

> General comment

The paper by Amos et al. presents a study based on satellite data and focusing on the role of mesoscale eddies in the transport of organic carbon from the nearshore region of the California Current System to the open ocean. The authors provide a quantification of the export flux sustained by the eddies and discuss the lateral isolation of mesoscale eddies with the use of model data and of a modelled passive tracer experiment.

The topic on which the authors focus is highly relevant and interesting. There are currently only a limited number of studies focusing on the role of mesoscale activity in the lateral transport of the organic transport, and therefore the study has potential to provide valuable information on the subject.

However, I have several fundamental concerns as regard to the methods and conclusions drawn by the authors on the basis of their analysis of the organic carbon concentration in the eddies. Also, the discussion of the results does not include an in depth analysis of the potential limitations of the adopted methods, both regarding the use of satellite data and regarding the parallel with the modelling study. A few choices in terms of analysis methods are not well justified. Furthermore, the authors forget to refer to some fundamental publications on the topic which already focused on the subject of eddy isolation and eddy organic carbon transport in upwelling regions, therefore lacking a thorough discussion of their results in the context of the present findings.

Unfortunately, I don't think that this study is yet ready for publication.

We are glad the reviewer considers our work highly relevant and interesting. We thank him/her for carefully reviewing our work and for providing very helpful suggestions and comments, which helped us substantially improve the manuscript. A detailed response to the comments and suggestions provided by the reviewer is presented below.

> Major comments

1. Biogeochemical fluxes are overlooked

The authors discuss the lateral eddy transport from the coast on distances of several hundreds of km. A typical eddy drifts at a speed of a few cm/s, meaning it takes months to cover a distance of 500 km, as confirmed by the authors at lines 84-85. Despite this time lag and the biological processes affecting POC concentrations, the authors use high particulate organic carbon (POC) concentrations as an indicator of the presence of coastal water in the eddies (page 4, lines 1-3), and state that the high organic carbon concentration of eddies found hundreds of km offshore proves that this organic carbon was trapped and transported from the coast to the open waters (page 4 lines 85-88). The authors forget to discuss thoroughly the substantial biogeochemical transformations that the organic carbon goes through in the course of a few months. There is

only a quick discussion of production in the eddies at lines 152-157, where the authors compare POC concentrations of nearshore and offshore generated eddies and use their difference to quantify the lateral transport of coastal organic carbon. This comparison overlooks the many differences between these two populations of structures found offshore, one younger (offshore generated eddies) and one older (nearshore generated eddies that drifted offshore in time). It also overlooks the potential differences in biogeochemical properties such as POC, nutrient concentrations and rates of production due to the different regions of formation, and finally overlooks the simple fact that the coastal carbon in older eddies may have sunk while they were drifting offshore. I therefore think that this comparison used to calculate the portion of coastal carbon that reaches the offshore region is not solid.

We appreciate and agree with the reviewer's comments. The high concentration of POC offshore associated with cyclonic eddies generated inshore could be due to trapping of POC, but also due to trapping of nutrients (the trapped nutrients could then lead to production inside the eddies as they propagate offshore and could contribute to additional POC). The carbon trapped in the coastal region may have also been locally remineralized and recycled into new carbon as the eddies propagate offshore. In all these scenarios, however, the POC enrichment in offshore waters would be driven by upwelled coastal water rich in POC/nutrients being trapped and transported offshore by the cyclones, which is our main point for this study. We clarify this in the text. We have also revised the text and now refer to "POC enrichment" offshore, rather than "POC transport". We clarified that the POC enrichment may be associated with direct lateral transport of POC, but also with offshore transport of nutrients sustaining local production and/or carbon being remineralized and recycle into new carbon.

We also agree with the reviewer that the trapped POC may sink out of the eddies over time and this is an uncertainty in our calculations. To the extent that this is true, then the lateral carbon transport should be higher than reported here (since part of the POC content of eddies generated inshore and located offshore would have been lost due to sinking). So with respect to the sinking particles, our estimates provide a lower-bound estimate of the POC enrichment offshore. That is one of the reasons we now refer to POC enrichment, rather than transport (i.e., POC that sinks out of an eddy is not captured by our method and also does not contribute to POC enrichment inside the eddy in offshore regions). We thank the reviewer for raising this important issue.

It would have actually been interesting, if not necessary, to know the mean properties and number of the onshore and offshore generated eddies that were analysed, and to see the equations used to calculate this net transport, as at the present moment the mathematical details of this calculation are not clear to me.

We thank the reviewer for this suggestion. We have added a table to the supplementary information with the properties of the eddies analyzed in the study. We have also added the equations used to calculate the volume transport and POC enrichment to the methods section.

As regard to the biogeochemical transformations of POC in drifting eddies, there are different pathways through which the organic carbon trapped at the coast may not reach the offshore waters, while the eddy is nevertheless showing a high POC signature. I strongly suggest that the authors better consider and discuss these possibilities and their implications for their study.

Recycling: the carbon trapped in the coastal region may have been locally remineralized and recycled into new carbon. Vertical export and new production: the organic carbon trapped at the coast may have sunk, while the organic carbon visible offshore may have been produced on the way to the open ocean through the utilization of the nutrients trapped at formation. The paper lacks a satisfactory discussion of these biogeochemical transformation and the conclusions may be affected by this shortcoming.

We agree with the reviewer that other processes related to trapping of coastal water may contribute to elevated concentrations of POC inside the cyclonic eddies generated near the coast compared to the eddies generated offshore. We have added a more thorough discussion to the text to clarify that the POC enrichment inside the cyclonic eddies located offshore that were generated inshore may be due to trapping POC at formation, as well as through recycling of the trapped POC into new carbon and local production inside the eddies through the utilization of nutrients trapped at formation. All of these mechanisms are related to eddies trapping and transporting coastal water offshore. We thank the reviewer for this important suggestion, which allowed us to better describe how trapping of coastal water by nonlinear cyclonic eddies can affect POC distribution in offshore waters. We have also revised the title of the manuscript to reflect this.

In order to overcome these limitations, the authors could use a different approach to the problem. The authors could change perspective and focus on the quantification of the eddy lateral transport of POC, compared to the total lateral transport (eddy and non eddy) of the same tracer in the region, without assuming that the POC found in the eddies was produced in the nearshore. This would still be a highly relevant quantification of the eddy lateral fluxes and of their potential contribution to the organic carbon cycle in the region. It is nearly impossible to determine where the carbon found in the eddies offshore was really produced near the coast, given these data. Such a change of focus would allow the authors to provide a much more solid quantification of the role of the eddies in the organic carbon lateral relocation, without the need of uncertain assumptions.

We appreciate the reviewer's suggestion, however we do not have a method for computing the non-eddy portion of the lateral transport based on observations. We are computing the POC enrichment associated with eddy-driven lateral transport of coastal water (the first part of the reviewer's suggestion), but we cannot compute the non-eddy component without a model. Other studies have focused on modeling the lateral transport by eddies and filaments (e.g. Gruber et al., 2011; Nagai et al., 2015; Lovecchio et al., 2018), but this is the first time this is being done with large-scale observations. We agree with the reviewer that it is difficult to determine where the carbon inside the eddies was really produced. As discussed above in our response to a previous comment by the reviewer, the POC enrichment inside eddies generated near the coast that propagated offshore could be due to trapping of POC during formation, but also due to trapping nutrients during formation which leads to production inside the eddy as it propagates offshore, as well as recycling of the trapped carbon. These mechanisms however still result from carbon and nutrients being trapped near the coast during formation. We agree with the reviewer that this was not clear in the original version of the manuscript, and we have revised the text accordingly.

2. Concern regarding the use of satellite data

The method used by the authors for the quantification of the organic carbon content of the eddies is based on the use of satellite POC data combined with the global eddy database made available online by Chelton et al. The idea is straightforward, and in line with several other previous studies of the eddy properties. However, a few recent studies have raised concern regarding the use of satellite data for the analysis of mesoscale activity. In particular, the paper by Amores et al. (2018) titled “Up to what extent can we characterize ocean eddies using present-day gridded altimetric products?” highlights an important underestimation in the number of eddies and most of all significant distortions in the identified eddies which result in the identification of fake large eddies. I strongly suggest to add the discussion of the uncertainties connected to such potential errors in the paper and possibly provide a rough estimate of the error on the flux quantification.

We again agree with the reviewer. The eddy dataset does likely underestimate the total number of eddies in the region given the resolution. A limitation discussed in Chelton et al. (2011) is that the sea level anomaly fields that are used to detect and track the eddies cannot resolve features with radius less than 40 km, therefore submesoscale and smaller mesoscale features are not accounted for. This may also lead to distortions in the identified eddies. The volume transport and POC enrichment calculations reported here are possibly underestimates since smaller eddies not captured in the data set are also certainly contributing to the offshore transport. With the upcoming launch of SWOT (Surface Water and Ocean Topography), which will provide SSH measurements down to a few km, we will possibly be able to address this issue in the future by including smaller eddies in the analyses and by reducing distortions in the characterization of the larger eddies. We followed the reviewer’s suggestion and added a discussion to the text and to the methods section about the limitations and uncertainties associated with the eddy dataset as discussed by Amores et al. (2018).

3. Concern regarding the model-based quantification of trapping in eddies

Part of the analysis is built on a quantification of the trapping and transport capacity of cyclonic eddies obtained through the use of some rather high resolution modelling study. This quantification of trapping in the eddies is made by releasing a passive tracer inside of the structures and calculating the portion of this tracer that is retained by the structure in time. However, I have doubts regarding the applicability of these results in the present study. First of all, there is not much information about the size of the sample of modelled cyclonic eddies that was studied. Also, with such a high resolution grid, possibly resolving small eddies of the size of a few tens of km, I wonder if the modelled eddy population compares to the eddy population identified in satellite data, which typically includes only large eddies. How many cyclonic eddies were studied? What is the size distribution of these eddies, their non-linearity parameter, and how does it compare to that of the eddies identified from the satellite?

Another, maybe more serious, concern regards the properties of the passive tracer released in these modelled eddies, which is not well illustrated in the paper. Does this passive tracer sink? This may be a very critical issue in terms of applicability of the model results, as POC (differently from dissolved tracers) can sink out of the eddies at speeds between 1 m/day to a above 100 m/day, therefore escaping from the first 400 m of depth pretty quickly. Also, the authors study the tracer that remains trapped in the first 400 m of depth, but they don’t specify if the tracer was released only in the surface of the eddy or also at depth. I believe that more details

regarding the experiments must be provided in order to assess whether if the resulting percentage of retained tracer can be used for the study of a sinking tracer such as POC, which is largely trapped and produced also below the surface.

We have revised the text to clarify this. We emphasize that the model results are *not* used to estimate POC enrichment in offshore regions. That calculation is entirely based on satellite observations, as described in Eq. 1 (equation has been added to the methods section).

The model results and tracer distribution are *only* used to estimate the “trapping efficiency” in Eq. 2 (equation has been added to the methods section) to obtain an estimate of the eddy-driven transport of water. Some studies in the past have estimated the volume transport by eddies by assuming that all the water is trapped and remains inside the eddies (i.e., assuming a trapping efficiency of 100%; e.g., Castelao, 2014), but we thought it would be better to get a rough estimate of the trapping efficiency since it is likely that not all water remains trapped. We estimate ~1 Sv of transport assuming that 65% of the water remains trapped inside the eddies (with the trapping efficiency of 65% coming from the tracer experiment). If we had used a trapping efficiency of 55% or 75%, we would get transports of 0.9 or 1.2 Sv, respectively. The goal of the volume transport calculation was simply to demonstrate that the eddy-driven transport of water is comparable to the transport of water in the upwelling jet. So the fact that our POC enrichment estimates, which are independent of the model, reveal a substantial lateral flux is consistent with that. This indicates that the lateral transport of carbon, nutrients, and other properties of the recently upwelled water by eddies is important. We added more details about the modelling and passive tracer experiment to the methods, thanks for pointing out that those were missing.

4. Eddy finding & trapping algorithm choice and performance

I would have appreciated to find some essential information (at least in the methods section) on the type of algorithm originally used by Chelton et al. for finding & tracking the eddies, as the entire study is based on the use of Chelton et al.’s eddy database. The authors could summarize some essential information about this eddy finding & tracking algorithm and of its performance, as their results may be sensitive to the algorithm’s ability to correctly identify and track the eddies. For example, the distinction between coastally- and offshore-generated eddies is sensitive to the presence of broken tracks that seem to originate offshore but are actually parts of longer tracks generated near the coast. This is generally carefully checked by observing an animation of the tracks on SSH. Was this checked by the authors?

We thank the reviewer for the suggestion and we added a summary of the Chelton et al. (2011) and Schlax and Chelton (2016) eddy detection and tracking algorithm and a discussion of the performance of this algorithm to the methods section. During our analyses we did check animations of the eddy tracks on SSH fields and observed that the algorithm does a good job detecting and tracking eddies in this region. The issue of broken tracks or the tracks of two eddies being merged into one did occur sporadically, but it was not a common occurrence so we proceeded with using the dataset.

We note that in an instance when an eddy is generated near the coast and propagates offshore and the eddy track is broken, the “new eddy” (beginning on the second half of the broken track)

will be mistakenly tagged as being generated offshore, when in reality it was generated inshore. This results in an underestimation of our estimate of POC enrichment offshore due to eddy transport. This is because the eddy mistakenly labeled as being offshore-generated will have more POC in its interior compared to eddies “truly” generated offshore (since that eddy was actually generated near the coast where the water is enriched in POC/nutrients). As a result, the composite POC content for offshore eddies would be artificially inflated, resulting in an underestimation of the POC enrichment (Eq. 1 in the manuscript). Thus, if the algorithm was perfect and there were no tracks broken, the estimate of POC enrichment due to eddy trapping would be larger than reported here. We expect this difference to be quite small, however, given that the analysis of the animations described above reveal that the fraction of broken tracks is small.

In general, eddies are generated by flow instabilities (coastal shear, currents hitting obstacles...) or wind stress curl injection, therefore I would expect less eddies to be generated away from the coast. Of these, many of them may be detaching from long coastal filaments and therefore could still trap coastal water (filaments often exceed 300 km of offshore length, especially in the California Upwelling). How many tracks are generated offshore of 300 km, and what mechanisms of generation are responsible for their formation?

We agree that some of the eddies may be detaching from coastal filaments that extend farther than 300 km from the coast and are therefore trapping the upwelled coastal water. From the eddy dataset, in the CCS between 33°-43°N there are a total of 369 cyclones and 410 anticyclones generated between 0-600 km from the coast between 1993-2015. 121 of the 369 cyclones and 135 of the 418 anticyclones are generated offshore between 300-600 km from the coast and the remaining eddies are generated between 0-300 km (this information has been added to Supplementary Table 1). We have not investigated the mechanisms that are responsible for the generation of the offshore eddies, but we suspect that interactions with the wind stress curl field are likely important.

In the scenario mentioned by the reviewer, if an eddy is detaching from a long coastal filament offshore of 300 km and it is still trapping the coastal water, then it will have a larger POC content (since the coastal water is enriched in POC/nutrients) compared to eddies generated locally offshore that are not trapping coastal water. Therefore this creates a bias toward a composite with higher POC for the eddies generated offshore. If we could exclude these occurrences and only consider eddies generated offshore that did not trap coastal water due to long filaments, then the difference in the POC content in the composites between eddies generated inshore and offshore in Eq. 1 in the manuscript would be even larger. To the extent that this occurs, our calculation of lateral POC enrichment due to cyclones provides a lower bound estimate of the actual POC enrichment.

Also, the authors select only the highly non-linear eddies from the entire eddy database (page 11, lines 236-240). However, since the authors use eddy tracks, it is not clear how are the slowly rotating eddies removed from the downloaded tracks, as the same eddy may speed up and slow down along its path. Do the authors use only portions of the tracks in which the eddy is sometime not highly nonlinear? How many tracks are studied in total? More information on this part of the methods may be useful to the reader.

The eddy dataset includes all mesoscale eddies, however Chelton et al. (2011) found that virtually all eddies at latitudes greater than 25° are nonlinear. For our analyses, we selected all eddies within the domain *that had sufficient POC satellite coverage*. We calculated the nonlinearity of those eddies and found that the majority of them were nonlinear for at least 80% of their lifetime. In total, 553 cyclonic eddy tracks were studied. These eddies were located in the CCS between 33°-43°N and 0-1500 km offshore during 1997-2010 (time period of satellite POC data). We thank the reviewer for this suggestion and we have added more information to the methods section and to Supplementary Table 1.

5. Choice of the nearshore/offshore domains

The authors identify the 300 km offshore line as the boundary between “nearshore” and “offshore” waters for the track generation. However, it is not clear to me why 300 km offshore should represent an important boundary. Is there a physical or biogeochemical divide between the region closer and further than 300 km from the coast? At page 11, lines 242-244 the authors quickly state that their results would not change if the boundary was instead placed at 350 km offshore. However, there is no motivation to support this other potential choice of offshore distance either. The authors should justify their choices better. Why are the first 300 km from the coast the “nearshore” region? Why not 200 km offshore or any other range of distances?

We chose 300 km as the offshore threshold to distinguish between nearshore and offshore because of the average width of the coastal band with high POC concentration. From the POC fields, we observed that a band of high POC concentrations is generally located within ~250 km from the coast (Figure C.1). Since we are using POC as a tracer of the upwelled coastal water, we chose a distance larger than the average width of the band with high POC to distinguish between the nearshore upwelled water and the offshore water. Also, the average offshore extent of frontal activity and the upwelling jet is about 300 km (Wang et al., 2015 estimate it at 304 km). If we had chosen 200 km as the threshold, then many eddies generated offshore of 200 km would still be able to trap water from inshore of the upwelling front in their interiors. We added more information about our choice of the 300 km threshold to the methods section. Thanks for this suggestion.

Figure C.1 – (a) Long-term average of POC concentrations for May-August. (b) POC concentration for 14 Aug 2007. Black line marks 300 km from the coastline.

6. It is not clear what the authors mean by “export”

The authors focus on the lateral transport of organic carbon by the eddies from the coast to the open waters, and mention the “total export” by the eddies (e.g., abstract, line 33). However, the word export is generally used to refer to the vertical relocation of the organic carbon from the euphotic layer (or mixed layer) to below. Do the author always refer to the lateral transport only? If so, I’d suggest to always add the adjective “lateral” as in “lateral export”, or to use expressions such as “lateral transport” or “lateral redistribution”. Otherwise, I’d suggest to clarify better when the study refers to lateral transport, and when to vertical export.

We thank the reviewer for raising this important issue, which we agree was not clear in the text. We have edited the text to refer specifically to lateral transport to distinguish from vertical transport/export. We also changed the notation and refer to POC enrichment rather than POC transport in most instances, because as mentioned by the reviewer the POC enrichment offshore may be due to POC transport, but also to transport of nutrients that may be used to sustain production inside the eddy and to recycling of the carbon trapped by the eddy.

Also, the authors talk about export from 0-300km offshore to 300km-900km offshore. It is not clear to me what a lateral export into an offshore box represents mathematically, as the lateral export is essentially a flux, and, as such, it is calculated as “concentration*velocity*surface” (and not *volume). I would rather talk about lateral export through the 300 km boundary, and not into the 300km-900km box. I am not sure if I understand the meaning of this flux, unless it is a vertical export. How is this export calculated? Could the author refer explicitly to the used mathematical method or equations?

The reviewer is correct – we are referring to the lateral transport across the 300 km boundary. We have revised the text to clarify this. We have also added the equation used to calculate the POC enrichment in offshore areas due to eddies trapping coastal water as they propagate offshore (Eq. 1) to the methods section of the manuscript.

7. Eddies or cyclonic eddies?

The authors essentially talk about cyclonic eddies rather than all the eddies, as one would expect from the paper’s title. In fact, they even move the anticyclonic eddy plots into the supplementary material. Even though the anticyclonic eddies do not have such an important role in the transport of POC, I’d suggest not to “hide” those plots, or, alternatively, I’d change the paper title and abstract explicitly focusing on the cyclonic eddy population only.

We followed the reviewer’s suggestion and added the plots about anticyclonic eddies to the main text.

7. Missing discussion of recent relevant literature

The paper discusses the interesting question of the role of eddies in the organic carbon transport, however the authors do not refer to some recent publications on the topic, and therefore miss the opportunity to compare their results to those of these previous studies.

On the topic of the eddy lateral isolation and eddy vertical structure, the authors could compare their modelling results to observational studies of the eddy structure and lateral isolation, such

as:

- Amores et al. 2016 “Coherent mesoscale eddies in the North Atlantic subtropical gyre: 3-D structure and transport with application to the salinity maximum”

- Pegliasco et al. 2015 “Main eddy vertical structures observed in the four major Eastern Boundary Upwelling Systems”

A similar (but model-based) study to the present one was presented by:

- Lovecchio et al. in 2018 “Mesoscale contribution to the long-range offshore transport of organic carbon from the Canary Upwelling System to the open North Atlantic”.

This study quantifies the eddy and filament lateral transport of organic carbon from the nearshore region of the Canary Upwelling system. How do your results compare to this study?

Further relevant studies are:

- Capet et al. 2013 “Eddies in Eastern Boundary Subtropical Upwelling Systems”

- Chaigneau et al. 2009 “Eddy activity in the four major upwelling systems from satellite altimetry”

We thank the reviewer for bringing our attention to these papers. We have included them in the text, which helped strengthen our discussion. It also helped highlight the novel aspects of our study, being the first to provide quantitative estimates of the influence of eddy driven transport of coastal water on the distribution of POC offshore using an observational approach.

> Detailed comments

Page 2, line 43: Please, define what you mean by nonlinear (e.g., rotational speed/drift speed >1?). Also, I'd say “of these eddies”.

Done.

Page 4, line 75: Please, state explicitly that you are using Chelton's database of already identified and tracked eddies. Also, I suggest to add more details about how these eddies were found and tracked by Chelton, in order to provide info on the used algorithm, its strengths and its limitations.

Done. We also added more information about Chelton et al. (2011) eddy detection and tracking algorithm to the methods section.

Page 4, lines 77-80: According to Chaigneau et al. 2009

(<https://doi.org/10.1016/j.poccean.2009.07.012>), eddies occupy about 24%-30% of the surface of upwelling regions, with the highest value found in the California Upwelling System. Why is the eddy surface fraction so low in the current study?

We believe our results are consistent with Chaigneau et al.'s results. In their study, the percentages of the surface area occupied by eddies includes both cyclones and anticyclones. Here, we calculated that 8.5-10.5% of the region 300-1200 km offshore is occupied by cyclonic eddies that were generated near the coast (Figure 2c in the manuscript) and 7% by those generated locally offshore (Figure 2a). Thus, about 16% of that area is occupied by cyclonic eddies. The anticyclonic eddies occupy about 14% of the region (Figure 2b,d). This gives us a

total of ~30% of the region that is occupied by eddies of either polarity, which is consistent with the value for the CCS presented in Chaigneau et al. (2009).

Page 4, lines 85-88: This sentence is too speculative and not well grounded. In 3-6 months POC may have gone through intense bgc transformations, it is not likely that the POC you find in an eddy months after its formation corresponds to the POC trapped at formation.

We have removed this sentence.

Page 5, lines 95-98: Please, do not put anticyclonic eddy plots in the supplementary, unless you focus your paper explicitly on cyclones. Also, differences in the water trapped by cyclones and anticyclones are not restricted to POC, but also include nutrients and living beings. I suggest the authors to keep these differences in mind, and consequently discuss the time evolution of the tracers in the eddies.

We have moved the figure about anticyclones to the main text, following the reviewer's suggestion. We have also reworded the text throughout to reflect that the trapped coastal water includes other properties in addition to POC.

Page 5, line 103 and Methods lines 206-221: POC anomalies are calculated as differences between the satellite-derived weekly mean POC field and the same field smoothed in space with a low pass filter. First off, there is no information about what are the properties of such a filter, how many points are averaged together in space, therefore results could be strongly influenced by the choice of the filter's properties. Second I find this method improper, as the POC value in each grid point is still influenced by the POC contained by a potential eddy in that point even after smoothing. This is not a valid reference field from which to calculate anomalies. With 13 years of satellite data available, the authors should use a valid reference mean as an annual mean POC field, or even better a monthly mean POC field interpolated weekly. I believe that the anomalies calculated with the use of a low pass filtered field as a reference field are not significant measures of the POC concentration in the eddies. Also, I don't believe that applying a gaussian filter to the calculated eddy anomalies is necessary, nor it is meaningful to regrid the mean eddy anomalies onto a higher resolution normalized grid, as no new information is generated through regridding of low resolution data.

We added information about the low pass filter (6° longitude by 6° latitude window) used to the methods section. Thanks for pointing out that this information was missing.

Following the reviewer's suggestion, we computed the long term means for the weekly POC fields and subtracted this from the original POC fields to compute the anomalies. The analysis using this method (removing long-term mean) yields similar results to the method used in the paper (removing low-pass filtered POC) (Figure C.2), demonstrating the robustness of the results.

We think that removing the long-term mean can introduce some errors in the analyses, however, especially near the upwelling front. Let's consider, for example, a simplified case of a region with a straight coast and an upwelling front parallel to the coast (i.e., 2D upwelling). If in a given

year upwelling favorable winds are stronger than average, then the upwelling front will be displaced farther offshore in comparison to the mean position of the front in other years. Since the area inshore of the front is characterized by high POC concentrations, removing the long-term mean POC concentration to compute the anomalies would result in a band of high POC in the region between the upwelling front for that year and the long-term mean position of the front. That band of high POC anomalies would be a result solely of the larger offshore displacement of the upwelling front in that year caused by anomalously strong upwelling winds. Thus, if an eddy were to be found in that band, it would be characterized by high POC anomalies that are partially due to the offshore displacement of the front due to winds, not due to eddy processes. We note that this is potentially an issue only in regions of strong gradients, especially near the front. If the background POC gradient is weak, then we would expect this issue to be less important.

Because of this, and considering that the results using the two methods are very similar (Figure C.2), we chose to use the spatial filter to isolate mesoscale variability, which has also been used in many previous studies (Chelton et al., 2004; Gaube et al., 2013; Gaube et al., 2014; Gaube et al., 2017; Yuan and Castelao, 2017, among many others). We note also that this is the procedure used to identify eddies in the Chelton data set [Schlax and Chelton (2016) say that they “spatially high-pass pre-filter the SSH fields to help the eddy identification procedure by removing large-scale signals in SSH that are unrelated to the mesoscale eddies that are of interest”]. We have added a sentence to the methods section clarifying this.

Original figures with spatial filter

Removing long term mean filter

Figure 4

Figure 4 – long term mean filter

Figure 5

Figure 5 – long term mean filter

Figure C.2 – Comparison of analyses described in the manuscript using two methods: On the left side, POC anomalies were calculated by removing large scale patterns via the use of a spatial filter. On the right, large scale patterns were determined by computing the long-term weekly averages. Note the similarity between results using the two different methods.

Regarding the comment about using a 2D Gaussian fit, it is very difficult to isolate the influence of an eddy from other mesoscale processes without that step. In the CCS, there is strong variability in POC associated with several mesoscale processes, such as eddies, the upwelling front, filaments, etc. There is overlap in the spatial scales of these processes – both eddies and the upwelling front scale with the internal Rossby radius (Kundu et al., 1975; Kamenkovich et al., 1986; Pedlosky, 1987; Cushman-Roisin and Tang, 1990), for example. If we use a filter that removes the signature of the upwelling front, it will also remove a lot of the eddy signature. In order to keep the eddy signature, then features associated with other mesoscale processes will still be present, at least partially. As an example, in Supplementary Figure 2c in the manuscript, the band of high POC values near the northeastern corner of the box is not related to the eddy, but these values would influence the composite averaging of the original POC without the Gaussian fitting. Computing the composite POC anomalies without the Gaussian fit (Figure C.3) produces results that are consistent with those presented in the manuscript (compare with Figure 3). Cyclonic eddies generated inshore that are still located inshore have the highest POC anomalies (Figure C.3a) as before. Cyclonic eddies generated inshore that are located offshore have higher concentrations of POC on average than eddies generated locally offshore (Figure C.3b,c), which is consistent with the results obtained when the 2D Gaussian fit is applied (Figure 3d in the manuscript). However, the composites of the original POC anomalies (without the Gaussian fit) show patches of POC along the edges of the 2 by 2 box that are not related to the eddies (upper right corner of Figure C.3a-c; this is consistent with Supplementary Figure 2b-c in the manuscript). Thus, using the composites without the Gaussian fit can lead to inaccurate results. After extensive testing, we find that applying the 2D Gaussian fitting to isolate the POC signature associated with eddies was a necessary step. We have included information about this in the text. We have also removed the step about regridding the data to higher resolution from our method.

Figure C.3 – (a-c) Composites of original POC anomalies (*without* applying 2D Gaussian fit) within 2 by 2 eddy radii for cyclonic eddies with amplitudes 9-11 cm. (d) Mean POC anomaly (mg m⁻³) and standard error calculated *within 2 by 2 eddy radius* grouped by eddy amplitude.

Note that this is different than in Figure 3d in the manuscript, where POC anomaly was computed within 1 eddy radius.

Note that the magnitude of the POC anomalies in Fig. C.3 is different than in Figure 3 in the manuscript, however, because here they were computed over the entire 2 x 2 radii box (instead of within 1 eddy radius), since the POC anomalies and the eddy centroid are not always coincident (note also that this can lead to negative values, if a large negative anomaly is located inside the 2 x 2 box).

Page 5, line 104: Why at 300 km from the coast and not 100 km or 200 km or any other distance? How do you define this boundary and what is its relevance? (same for Methods, lines 242-244)

Added more information to the “Mesoscale Eddies” section in the Methods about our reasoning for using 300 km as the boundary. See also response to major comment 5 above.

Page 5, line 110: Analogously, why 300 – 600 km offshore?

We simply used a similar band width as 0-300 km. Also, this is a useful width because it allows for a good number of eddies to be included in the composites. If the offshore band was too narrow then we would end up having few eddies in which cloud-free POC data were available. We have added information about this to the “Mesoscale Eddies” section in the Methods.

Page 5, lines 110-111: The high POC concentration found offshore in eddies generated near the coast does not indicate that the carbon trapped at formation is still in the eddies many months later. That carbon may have completely sunk, and have been substituted by newly generated carbon through the use of trapped nutrients.

Yes, we agree with the reviewer that the high POC concentration found in eddies generated near the coast when they are located offshore may not be just from trapping POC near the coast. During formation, eddies may be trapping carbon and transporting it offshore, or trapping nutrients and transporting that offshore. As the eddies propagate offshore, the trapped nutrients may be utilized for new production. The POC trapped near the coast may also have been locally remineralized and recycled into new carbon, as the reviewer pointed out. In all these cases, however, the POC enrichment offshore is related to trapping of coastal water by the eddy and subsequent offshore lateral transport, which is our main point. We thank the reviewer for pointing out that this was not clear and we have edited the text to address this.

Page 5, line 114: Why now 900-1200km offshore, and what happens in the 600-900 km region that is never mentioned? I suggest to clearly define these boundaries before discussing the results or in the methods and to justify their choice, as above. Also, it would be useful to associate these distances to the time taken by the eddies to reach them, for example on average in the entire eddy track population.

The goal of this analysis was to identify the width of the coastal band in which POC enrichment was observed associated with the lateral transport of coastal water by eddies. We used regions with similar widths as the bands we used before, 0-300 and 300-600 km. To extend the analyses farther offshore, we used 600-900, 900-1200 and 1200-1500 km. Offshore of 1200 km, we can

no longer see a difference in the POC content between eddies generated inshore and generated offshore (Figure 4 in the manuscript). We have added a discussion to the text about why we chose these regions for the analyses. We also added information about the average time it takes for the eddies to reach the offshore region, thanks for the suggestion.

Page 6, lines 118-119: This is too speculative. POC could also be remineralized, as well as it may sink. It is simply not possible to assess the biogeochemical transformations from satellite data. I suggest either to discuss all the possibilities and their consequences for the results, or not to attempt to pick one single explanation for the POC tendency. Also, the fact that POC decreases in time is expected.

We have removed this from the text.

Page 6, lines 128-129: This sentence is unclear, please rephrase it.

Done.

Page 6, line 134: Too speculative. Also, how many eddies are really generated offshore of 300 km from the coast and by which mechanisms? Are they not just pinching from long filaments? (which would indicate that they also trap some coastal water) Are there islands that perturb the flow? What is the seasonality of this offshore eddy generation? Again, what is the significance of distinguishing between eddies generated closer and farther than 300 km from the coast?

We have removed the sentence that the reviewer considered too speculative.

We have added information about number of eddies generated offshore to Supplementary Table 1 in the manuscript. As for the question about the mechanism(s) that are responsible for the generation of the offshore eddies, please see our response to the reviewer's major comment 4 above (paragraph starting with "*We agree that some of the eddies may be detaching from coastal filaments...*").

We chose 300 km as the offshore threshold to distinguish between nearshore and offshore because of the average width of the band of the high concentration of POC (please see our response to the reviewer's major comment 5 above and Figure C.1).

Page 6, line 138: 6 cyclones per year sounds a bit low. Is this because you are only looking at a subset of the entire eddy tracks population?

Yes, the reviewer is correct. We are only looking at eddies that are generated inshore of 300 km *and* that propagate offshore of 300 km. Almost half of the cyclonic eddies that are formed inshore of 300 km do *not* propagate offshore of 300 km from the coast. Also, only eddies with lifetimes longer than 4 weeks are included in the analyses since the eddy dataset excludes eddies with shorter lifespans. We edited the text to clarify that the analysis shown in Figure 6a only includes cyclones with lifetimes longer than 4 weeks that propagate offshore of 300 km.

Page 7, line 140: In your modelling study, did you restrict your analysis to large eddies comparable to those found in satellite data?

Yes, the cyclonic eddies selected in the model for the tracer experiment were of similar size and amplitude to the large satellite-detected eddies. We added this information to the text.

Page 7, lines 146-150: Why do you need in situ POC when you have the vertically integrated POC from satellite data already available, which would allow you to make explicitly the same flux calculation? How many in situ measurements do you use per each eddy?

We use the in situ POC to determine a relationship between the surface POC and POC integrated between the surface and 100 m, following the same methodology used by Allison et al. (2010). The satellite-derived POC provides concentrations at the surface. We use the relationship between the surface and the vertically integrated POC determined from the in situ data to estimate the vertically integrated POC from the satellite-derived concentrations. This is essentially the same methodology used in global products of integrated POC (e.g. Duforêt-Gaurier et al., 2010) using in situ profiles from different parts of the ocean. For our study, we are using in situ data collected in the CCS, so the relationship between the surface and integrated POC content is especially calibrated for our area of interest.

Page 7, lines 152-157: This approach is not solid, as discussed above in the major comments

This section has been re-written to better explain the approach as well as its implications and limitations.

Page 8, lines 169-170: As the authors state, filaments extend easily up to 400 km offshore. Eddies often detach from these filaments, trapping coastal upwelled water that is transported far offshore by the filaments. Therefore, again, distinguishing between eddies generated at offshore distances smaller and larger than 300 km offshore does not necessarily separate eddies that are influenced by upwelled waters and eddies that are not.

We agree with the reviewer, as discussed above in our response to major comment 4. In occurrences where an eddy is detaching from a long coastal filament offshore of 300 km, it is trapping the coastal water and will have a larger POC signature (since the coastal water is enriched in POC) than an eddy generated locally offshore without the influence of a long filament. This creates a bias toward a composite with higher POC for the eddies generated offshore, decreasing the difference in POC content between eddies generated inshore that propagate offshore and those generated offshore. If we could separate out the eddies that are detaching from long filaments (> 300 km) from those generated offshore without the influence of the filaments, then the difference in the POC content in the composites between eddies generated inshore and offshore in Eq. 1 in the manuscript would be larger. In that case, the process we are describing would become even clearer and the significance of the trapping and transporting mechanisms would be even larger. More details about this are provided in our response to major comment 4.

Page 9, lines 184-186: The high POC concentration and potential remineralization of the eddy POC could also lead to degassing of CO₂ from the eddies. The discussion is just too speculative and sounds out of context.

We agree with the reviewer, and as such we have removed this from the text.

As a final comment, we note that in the original version of the manuscript, the distance of 300 km from the coast had been incorrectly calculated by simply computing that distance in the zonal direction. In the revised manuscript, the distance of 300 km is calculated in the direction perpendicular to the coastline. Although the difference between the two methods is small in many regions, it is actually quite large near Point Conception (Supplementary Figure 1). As a result, many of the eddies generated within 300 km from the coast in that region had been mistakenly assigned as being generated in the offshore region in the original manuscript. That artificially increased the POC content of eddies generated offshore, resulting in a smaller estimate of POC enrichment due to eddy transport (according to Eq. 1 in the manuscript). Using the correct calculation of the distance resulted in slightly different POC values and estimates for the volume transport and POC enrichment, however it did not affect the conclusions. This has been fixed in this revised version. We apologize for the confusion.

We thank again the reviewer for the very helpful comments. We realize that providing detailed comments like these is time consuming, so we are especially appreciative of the effort.

Reviewers' comments:

Reviewer #1 (Remarks to the Author):

I thank the authors for addressing my comments and concerns. Their use of the Gaussian fit is now much clearer and I agree that it is appropriate. Note that in your R2R you computed time series from 2x2 boxes. This is not advised, we usually only include observations from within a circle with radius of the eddy centered on the SLA extremum or eddy centroid. Nevertheless, your use of the Gaussian fit takes care of this issue.

Reviewer #2 (Remarks to the Author):

Overall, the authors have done a very good job of improving their manuscript and clarifying my own and other reviewer's comments. I like the paper very much. Were this a review for JGR, Oceans, or Journal of Marine Systems, I would be done and congratulate the authors on a very good paper. But I stand by my original comments that this paper represents incremental advancement and is more appropriate for a specialized oceanographic journal than in Nature Communications. I acknowledge that my bar may be too high here and respect that publishing this in Nature is an editorial decision.

To help the editor understand what is new here and what is not, I highlight the main points of the paper from their abstract:

1) "Here, we use satellite-derived measurements of particulate organic carbon (POC) as a tracer of coastal water to show that cyclonic eddies located offshore that were generated near the coast contain higher carbon concentrations in their interior than eddies of the same amplitude generated locally offshore."

New: estimates of POC transport from satellite-derived measurements.

Not new: that cyclonic eddies generated nearshore have higher concentrations of upwelled quantities than offshore generated eddies.

There have been excellent modeling studies that show similar trapping of nearshore properties. The authors are aware of the Chenillat paper on this subject, but they do not seem to be aware of Combes et al., 2013: "Cross-shore transport variability in the California Current: Ekman upwelling vs. eddy dynamics." In that study, authors use passive tracers as proxies for nearshore upwelled biogeochemical fields and show a significant asymmetry in transport associated with cyclonic vs anti-cyclonic eddies (e.g., their figure 7).

The authors appropriately point out in their rebuttal that their study is observational and not based on modeling as in Combes or Chenillat. This fact is meaningful, but the basic result itself is just not surprising.

2) "This indicates that eddies are in fact trapping and transporting coastal water that is rich in carbon and nutrients offshore, resulting in a POC enrichment of 20.9 ± 11 Gg year⁻¹ in the offshore region."

New: The number 20.9 ± 11 Gg/year is indeed a new result, and represents an important advance for the field.

Not new: eddies are in fact trapping and transporting coastal water that is rich in carbon and nutrients offshore. Comments to point (1) apply here.

3) "Eddy-driven transport extends for 1000 km offshore, which is wider than the region typically influenced by upwelling jet separation and filaments. "

Not new: the whole sentence.

The references given in my first review going back over 20 years all show this: Kelly et al. (1998), Marchesiello et al (2003), and Sotka et al. (2004). Combes et al. (2013) is another good reference for this. In their rebuttal, the authors do not disagree with this point, yet they kept the sentence in the abstract.

If the editor chooses to publish, I suggest that the title be changed.

The original title was: "Offshore transport of carbon in the California Current System by mesoscale eddies"

My original comment was as follows:

9) The authors make the appropriate point (line 180) that the total carbon export is probably much higher than they can state through their estimate because dissolved concentrations can be 10-25

times higher than POC. This uncertainty calls into question the significance of the calculation overall and even the title of the paper, which implies it's a more comprehensive estimate.

Based on that, the authors resubmitted a revision titled: "Offshore transport of coastal upwelled water in the California Current System by mesoscale eddies" which is even more comprehensive a statement than their original title.

I suggest:

"Offshore transport of particulate organic carbon in the California Current System by mesoscale eddies"

Reviewer #3 (Remarks to the Author):

Review of "Offshore transport of coastal upwelled water in the California Current System by mesoscale eddies" by Amos et al.

General comment

After having re-read the manuscript and the entire set of reviewers' comments and authors' rebuttal, I appreciate the authors' effort put in writing a long and detailed rebuttal to the many questions and comments received but I still find that the paper needs further work before being ready for publication.

The results are not presented in a very logical way, with pieces of information being hidden in paragraphs that mostly focus about something else and some calculations not being justified by the authors. Sometimes it seems that numbers and estimates are presented without a reason (see lines 158-159, lines 177-179, lines 267-268), the authors should make a better effort at explaining where these numbers they use come from. I am still not convinced that using the difference between nearshore-generated eddies POC content, and the tracer in offshore generated eddies is an objective measure of the POC transport to offshore regions, as eddy physical/biogeochemical properties can be very different given the different formation mechanisms. I think that the use of a Gaussian filter may be avoided choosing a better reference average for the POC field.

I also fundamentally agree with the suggestion of Reviewer nr.2, who states that the paper mostly extends the analysis already made by previous studies and that it should probably be addressed to a topical journal focusing on oceanography, rather than Nature Communications. Even though the subject is relevant, many of the results and explanations presented in the paper are pretty obvious given the existing literature on the subject of mesoscale eddies (see for example: lines 96-104, 128-130), or are extremely specific to the region of study (see lines 131-138). As regard to the main

subject of the study, I can mention other publications having explored the topic of the organic carbon transport by mesoscale eddies, even though with a modelling approach. Even though the applied methods are different, the eddy lateral transport for longer than 400 km offshore was shown in the same region by Nagai et al. 2015 (<https://doi.org/10.1002/2015JC010889>). For the Canary Current System Lovecchio et al. 2018 (<https://doi.org/10.5194/bg-15-5061-2018>) demonstrate that the eddy transport of organic carbon extends up to 2000 km off the coast. Given these previous studies, the lateral transport of organic material by mesoscale eddies is expected in all the upwelling regions. A comparative study between the organic offshore carbon transport in the 4 major upwelling systems (which is surely possible using satellite data) or some other generalization of the flux impact would have sounded much more suitable for a journal of general interest such as Nature Communications.

Detailed comments

1. The authors still don't state in the text why they focus on the 300 km – 1200 km region, when they first introduce it, and the range sounds . Why not add on page 4 line 82 a short explanation such as: “where the 300 km nearshore band is the most POC rich region”, or similar
2. The title at line 106 “lateral transport of coastal water” should probably be of “coastal tracers” as the section is not about mass transport
3. I do not agree with the sentence at lines 127-128: “...indicating that the lateral transport of coastal water trapped by cyclonic eddies is most important within about 1000 km from the shore”. In fact, the authors refer to the organic carbon signature, which does not necessarily measure the trapping of coastal water. One would have to look at other tracers, such as salinity, spiciness or even nutrients to make this general statement about whether there is still coastal water in those eddy cores. I would rather talk about “lateral transport of organic carbon from the nearshore region to the open waters”, omitting the adjective “coastal”, as it is not given that the carbon found at 1000km offshore was generated at the coast, as already discussed.
4. Lines 158-159: “using different trapping efficiencies of 55% and 75%...” – where do these alternative values come from? Why is it significant to compare to these values, which are not even in the range of the estimated 65%±7% presented by the authors a few lines above? Are these numbers used just to get results that are closer to previous estimates? The authors should justify in the text why these percentages should be considered.
5. Lines 176-179: Why “shallower layers of 20 m and 50 m”? What is the significance of these depths? Do they represent anything physically or biologically?
6. Same problem at lines 267-268: what's the meaning of hypothesizing using 350 km offshore? Why should that number be relevant to the reader? I don't understand the authors' approach to these alternative hypotheses. Why not making a general statement instead of

comparing to some other rather subjective values? In my opinion, it is expected that there is no difference shifting the line by only 50 km.

7. Authors' rebuttal: figure A.1/C.3 – Being familiar with eddy composites, I am very surprised of seeing the upwelling POC rich band in the eddy anomalies, as the upwelling is a mean phenomenon that should be filtered away by subtracting the mean. This should work if the reference mean is chosen in a way that captures the recurrent regional variability in the system, eg. a mean that varies seasonally or monthly. Obviously, choosing a long-term annual mean would leave the upwelling signature there (thus requiring the gaussian filter to make the eddy pop out better), but it would likely also over/underestimate the actual POC anomaly in the eddies, as the background POC content surely varies seasonally too.

Reviewer #1 (Remarks to the Author):

I thank the authors for addressing my comments and concerns. Their use of the Gaussian fit is now much clearer and I agree that it is appropriate. Note that in your R2R you computed time series from 2x2 boxes. This is not advised, we usually only include observations from within a circle with radius of the eddy centered on the SLA extremum or eddy centroid. Nevertheless, your use of the Gaussian fit takes care of this issue.

We would like to thank the reviewer for his comments throughout the process, which helped us improve the manuscript substantially.

Reviewer #2 (Remarks to the Author):

Overall, the authors have done a very good job of improving their manuscript and clarifying my own and other reviewer's comments. I like the paper very much. Were this a review for JGR, Oceans, or Journal of Marine Systems, I would be done and congratulate the authors on a very good paper. But I stand by my original comments that this paper represents incremental advancement and is more appropriate for a specialized oceanographic journal than in Nature Communications. I acknowledge that my bar may be too high here and respect that publishing this in Nature is an editorial decision.

To help the editor understand what is new here and what is not, I highlight the main points of the paper from their abstract:

1) "Here, we use satellite-derived measurements of particulate organic carbon (POC) as a tracer of coastal water to show that cyclonic eddies located offshore that were generated near the coast contain higher carbon concentrations in their interior than eddies of the same amplitude generated locally offshore."

New: estimates of POC transport from satellite-derived measurements.

Not new: that cyclonic eddies generated nearshore have higher concentrations of upwelled quantities than offshore generated eddies.

There have been excellent modeling studies that show similar trapping of nearshore properties. The authors are aware of the Chenillat paper on this subject, but they do not seem to be aware of Combes et al., 2013: "Cross-shore transport variability in the California Current: Ekman upwelling vs. eddy dynamics." In that study, authors use passive tracers as proxies for nearshore upwelled biogeochemical fields and show a significant asymmetry in transport associated with cyclonic vs anti-cyclonic eddies (e.g., their figure 7).

The authors appropriately point out in their rebuttal that their study is observational and not based on modeling as in Combes or Chenillat. This fact is meaningful, but the basic result itself is just not surprising.

We are glad the reviewer liked the paper very much, and we thank him/her for the thoughtful comments. We have revised the text to address the additional comments listed here. We have added a citation for the modeling paper by Combes et al. (2013), thanks for the suggestion.

2) "This indicates that eddies are in fact trapping and transporting coastal water that is rich in carbon and nutrients offshore, resulting in a POC enrichment of 20.9 ± 11 Gg year⁻¹ in the offshore region."

New: The number 20.9 ± 11 Gg/year is indeed a new result, and represents an important advance for the field.

Not new: eddies are in fact trapping and transporting coastal water that is rich in carbon and nutrients offshore. Comments to point (1) apply here.

3) "Eddy-driven transport extends for 1000 km offshore, which is wider than the region typically influenced by upwelling jet separation and filaments. "

Not new: the whole sentence.

The references given in my first review going back over 20 years all show this: Kelly et al. (1998), Marchesiello et al (2003), and Sotka et al. (2004). Combes et al. (2013) is another good reference for this. In their rebuttal, the authors do not disagree with this point, yet they kept the sentence in the abstract.

We have revised the sentence referred to by the reviewer, which now reads "This offshore enrichment of POC due to the influence of coastally-generated eddies extends for 1000 km from shore...". We agree that this better reflects the novel contributions of our manuscript. As described in the previous round of revisions, the manuscripts by Kelly et al. (1998), etc., show that enhanced eddy activity extends a large distance from the coast, but these articles do not show that coastal tracers can be trapped and transported offshore by those eddies (e.g., linear eddies would result in enhanced eddy activity offshore but in no trapping and transporting of coastal water).

If the editor chooses to publish, I suggest that the title be changed.

The original title was: "Offshore transport of carbon in the California Current System by mesoscale eddies"

My original comment was as follows:

9) The authors make the appropriate point (line 180) that the total carbon export is probably much higher than they can state through their estimate because dissolved concentrations can be 10-25 times higher than POC. This uncertainty calls into question the significance of the calculation overall and even the title of the paper, which implies it's a more comprehensive estimate.

Based on that, the authors resubmitted a revision titled: "Offshore transport of coastal upwelled water in the California Current System by mesoscale eddies" which is even more comprehensive a statement than their original title.

I suggest:

"Offshore transport of particulate organic carbon in the California Current System by mesoscale eddies"

We appreciate the reviewer's suggestion to change the title. However, as explained throughout the paper, the enrichment of POC in the offshore region is likely due not only to the trapping and offshore transport of coastal POC, but also due to the trapping and redistribution of coastally upwelled nutrients which supports production as the eddies move offshore. We think that "coastal upwelled water" is more inclusive of the material that is being trapped near the coast and redistributed offshore by eddies.

Reviewer #3 (Remarks to the Author):

Review of "Offshore transport of coastal upwelled water in the California Current System by mesoscale eddies" by Amos et al.

General comment

After having re-read the manuscript and the entire set of reviewers' comments and authors' rebuttal, I appreciate the authors' effort put in writing a long and detailed rebuttal to the many questions and comments received but I still find that the paper needs further work before being ready for publication.

The results are not presented in a very logical way, with pieces of information being hidden in paragraphs that mostly focus about something else and some calculations not being justified by the authors. Sometimes it seems that numbers and estimates are presented without a reason (see lines 158-159, lines 177-179, lines 267-268), the authors should make a better effort at explaining where these numbers they use come from. I am still not convinced that using the difference between nearshore-generated eddies POC content, and the tracer in offshore generated eddies is an objective measure of the POC transport to offshore regions, as eddy physical/biogeochemical properties can be very different given the different formation mechanisms. I think that the use of a Gaussian filter may be avoided choosing a better reference average for the POC field.

I also fundamentally agree with the suggestion of Reviewer nr.2, who states that the paper mostly extends the analysis already made by previous studies and that it should probably be addressed to a topical journal focusing on oceanography, rather than Nature Communications. Even though the subject is relevant, many of the results and explanations presented in the paper are pretty obvious given the existing literature on the subject of mesoscale eddies (see for example: lines 96-104, 128-130), or are extremely specific to the region of study (see lines 131-138). As regard to the main subject of the study, I can mention other publications having explored the topic of the organic carbon transport by mesoscale eddies, even though with a modelling approach. Even though the applied methods are different, the eddy lateral transport for longer than 400 km offshore was shown in the same region by Nagai et al. 2015 (<https://doi.org/10.1002/2015JC010889>). For the Canary Current System Lovecchio et al. 2018 (<https://doi.org/10.5194/bg-15-5061-2018>) demonstrate that the eddy transport of organic carbon extends up to 2000 km off the coast. Given these previous studies, the lateral transport of organic material by mesoscale eddies is expected in all the upwelling regions. A comparative study between the organic offshore carbon transport in the 4 major

upwelling systems (which is surely possible using satellite data) or some other generalization of the flux impact would have sounded much more suitable for a journal of general interest such as Nature Communications.

We thank the reviewer for re-reading the manuscript and for providing additional comments and suggestions. We have revised the manuscript, providing additional justification for the calculations presented following the reviewer's recommendations. Although we agree that extending the work to the other eastern boundary current systems (EBCS) would be very interesting, the limiting factor is the availability of in situ data. We are not aware of similar depth-resolving in situ POC observations spanning multiple years existing in other upwelling systems. As data are gathered in the other three EBCS, however, we hope that our manuscript will provide guidance for additional studies. We expect that the process described here will be highly relevant in those other systems.

Detailed comments

1. The authors still don't state in the text why they focus on the 300 km – 1200 km region, when they first introduce it, and the range sounds . Why not add on page 4 line 82 a short explanation such as: “where the 300 km nearshore band is the most POC rich region”, or similar

We thank the reviewer for pointing this out. We added text to further explain why we focus on the 300-1200 km region.

2. The title at line 106 “lateral transport of coastal water” should probably be of “coastal tracers” as the section is not about mass transport

We thank the reviewer for this suggestion and have modified the section heading accordingly.

3. I do not agree with the sentence at lines 127-128: “...indicating that the lateral transport of coastal water trapped by cyclonic eddies is most important within about 1000 km from the shore”. In fact, the authors refer to the organic carbon signature, which does not necessarily measure the trapping of coastal water. One would have to look at other tracers, such as salinity, spiciness or even nutrients to make this general statement about whether there is still coastal water in those eddy cores. I would rather talk about “lateral transport of organic carbon from the nearshore region to the open waters”, omitting the adjective “coastal”, as it is not given that the carbon found at 1000km offshore was generated at the coast, as already discussed.

We agree with the reviewer that the sentence should be focused on the organic carbon signature. This sentence has been revised to emphasize that the POC enrichment in offshore waters associated with eddies that were generated near the coast and propagated offshore is most important within 1000 km from the coast. Thanks for pointing out that this was not clear.

4. Lines 158-159: “using different trapping efficiencies of 55% and 75%...” – where do these alternative values come from? Why is it significant to compare to these values, which are not even in the range of the estimated 65%+-7% presented by the authors a few lines above? Are

these numbers used just to get results that are closer to previous estimates? The authors should justify in the text why these percentages should be considered.

We used trapping efficiencies of 55% and 75% to provide error bounds for the volume transport estimate. We have revised these values according to the reviewer’s comment to $\pm 7\%$. We have added information to the text to explain this.

5. Lines 176-179: Why “shallower layers of 20 m and 50 m”? What is the significance of these depths? Do they represent anything physically or biologically?

This sentence was added in response to a comment by reviewer 1 in the previous round, who suggested that we put error bounds on our carbon estimate by making the estimates for a range of depths, not just 100 m. Additionally, we showed that the correlation coefficient between surface POC and the integrated POC for different depth ranges (top 10 m, top 20 m, top 30 m, etc.) is approximately depth independent (in the top 100 m; Figure C.1), justifying our approach.

Figure C.1 – Correlation coefficient between surface POC and integrated POC for different depth ranges (top 10 m, top 20 m, top 30 m, etc.).

6. Same problem at lines 267-268: what’s the meaning of hypothesizing using 350 km offshore? Why should that number be relevant to the reader? I don’t understand the authors’ approach to these alternative hypotheses. Why not making a general statement instead of comparing to some other rather subjective values? In my opinion, it is expected that there is no difference shifting the line by only 50 km.

The goal of using a different distance to distinguish between inshore and offshore was to demonstrate that our results and conclusions are not sensitive to the value chosen. As we mention in the manuscript, we chose a distance of 300 km because it is consistent with the average width

of the coastal band with high POC. We thought it would be important to indicate that using other values (e.g., 350 km) yields consistent results. That information was added to the text.

7. Authors' rebuttal: figure A.1/C.3 – Being familiar with eddy composites, I am very surprised of seeing the upwelling POC rich band in the eddy anomalies, as the upwelling is a mean phenomenon that should be filtered away by subtracting the mean. This should work if the reference mean is chosen in a way that captures the recurrent regional variability in the system, eg. a mean that varies seasonally or monthly. Obviously, choosing a long-term annual mean would leave the upwelling signature there (thus requiring the gaussian filter to make the eddy pop out better), but it would likely also over/underestimate the actual POC anomaly in the eddies, as the background POC content surely varies seasonally too.

Using a seasonal or monthly mean to capture and filter the upwelling signal will not result in the upwelling signature being completely filtered out. For example, the upwelling signature during August of a given year will be different than the average signature for all Augusts depending on wind conditions (e.g., following a strong upwelling-favorable wind event, the upwelling signature will be stronger and the front will be shifted offshore in comparison to the monthly average). Given a “weather-band” variability of 3-10 days, a seasonal or monthly mean will not fully remove these short term pulses in upwelling, resulting in the upwelling POC rich band to still be present in the composites. The same is true for the signature of filaments. If a filament is observed on a given day in August 2012, for example – removing the long-term average for all Augusts would not remove the signature of the filament in 2012.